METHODS AND RESOURCES

# Defining cellular diversity at the swine maternal–fetal interface using spatial transcriptomics and organoids

Cole R. McCutcheon[1], Allyson Caldwell[1], Liheng Yang[1], Elisa Crisci[2],
Jonathan Alex Pasternak[3], Carolyn B. Coyne [1,4]*

**1** Department of Integrative Immunobiology, Duke University School of Medicine, Durham, North Carolina, United States of America, **2** Department of Population Health and Pathobiology, College of Veterinary Medicine, North Carolina State University, Raleigh, North Carolina, United States of America, **3** Department of Animal Sciences, Purdue University, West Lafayette, Indiana, United States of America, **4** Duke Human Vaccine Institute, Durham, North Carolina, United States of America

* carolyn.coyne@duke.edu

## Abstract

The placenta is a dynamic, embryo-derived organ essential for fetal growth and development. While all eutherian mammals have placentas composed of fetal-derived trophoblasts that mediate maternal–fetal exchange, their anatomical and histological structures vary across species due to evolutionary divergence. Despite the cellular heterogeneity of porcine trophoblasts in vivo, understanding the mechanisms driving porcine placental development has been limited by the lack of in vitro models replicating this heterogeneity. In this study, we derived swine trophoblast organoids (sTOs) from full-term porcine placentas, retaining key transcriptional signatures of in vivo trophoblasts. To identify conserved cell populations, we integrated Visium spatial transcriptomics from mid-gestation porcine placentas with single-cell transcriptomics from sTOs. Spatial transcriptomics revealed novel markers of the porcine uterus and placenta, enabling precise separation of histological structures at the maternal–fetal interface. The integration of tissue and sTO transcriptomics showed that sTOs spontaneously differentiate into distinct trophoblast populations, with conserved gene expression and cell communication programs. These findings demonstrate that sTOs recapitulate porcine placental trophoblast populations, offering a powerful model for advancing placentation research. Our work also provides a spatially resolved whole-transcriptome dataset of the porcine maternal–fetal interface, opening new avenues for discoveries in placental development, evolution, and health across mammals.

which permits unrestricted use, distribution, and reproduction in any medium, provided the original author and source are credited.

**Data availability statement:** All code is publicly available on the Coyne Lab Github (https://github.com/CoyneLabDuke/Swine-maternal-fetal-interface) and archived via Zenodo (https://doi.org/10.5281/zenodo.15747602). All raw and mapped data obtained via Bulk RNA-Sequencing, Single Cell, and Visium Spatial Transcriptomics data generated in this paper are deposited on the Gene Expression Omnibus (GSE296975 and GSE301436) and additionally are available via the Sequencing Read Archive (under bioproject number PRJNA1252046). Additionally, a searchable, user-friendly version of the Visium Spatial Transcriptomics dataset has been uploaded at https://coynelab.shinyapps.io/myshinyapp/.

**Funding:** This work was supported by funds from Duke University (C.B.C.). Placental tissue samples were derived from work supported by the Foundational and Applied Science Program, project award no. 2023-67015-39338 (J.A.P.), from the U.S. Department of Agriculture's National Institute of Food and Agriculture (NIFA), https://nifa.usda.gov. The funders had no role in the study design, data collection and analysis, decision to publish, or preparation of the manuscript.

**Competing interests:** The authors have declared that no competing interests exist.

**Abbreviations:** BSA, Bovine Serum Albumin; CCDC162, coiled coil domain containing 162; CTBs, cytotrophoblasts; CTSB, cathepsin B; CTSL, cathepsin L1; DMEM, Dulbecco's Modified Eagle Medium; EVTs, extravillous trophoblasts; hTOs, human TOs; ISGs, interferon-stimulated genes; MMP7, matrix metalloprotease 7; NHP, non-human primate; OCT, optimal cutting temperature; PAG2L4, pregnancy-associated glycoprotein L4; PBS, Phosphate-Buffered Saline; PSGs, pregnancy-stimulated genes; RT, reverse transcription; SPINK4, serine peptidase inhibitor kazal type 4; SRA, Sequence Read Archive; STB, syncytiotrophoblast; sTOs, swine trophoblast organoids; tTOM, trophoblast organoid medium.

## Introduction

The defining feature of eutherian mammals is the presence of the fetal-derived placenta. The placenta is critical for fetal growth due to its numerous functions ranging from nutrient transfer [1,2], steroidogenesis [3], and immunological defense [4,5]. While these placental functions are largely maintained across species, the placenta has repeatedly evolved via a combination of convergent and divergent evolution resulting in variable placental morphologies across species [6,7].

Placental structure is commonly categorized by the tissue layers separating the maternal and fetal vasculature [7,8]. In the most invasive placental type, termed hemochorial, the placenta erodes the uterine epithelium and remodels the uterine vasculature resulting in the influx of maternal blood which directly bathes the placental epithelial cells, termed trophoblasts [7]. This type of placentation is observed in many species, including humans and rodents. Conversely, porcine placentae are considered the least invasive placental type, termed epitheliochorial. In this system, the uterine vasculature, stroma, and epithelium remain intact due to a lack of invasion by the trophoblast cells [7]. In this case, the primary contact point for the placenta is the uterine epithelium. Critically, regardless of the placental type or structure, all gas and nutrient transfer must occur across the trophoblast layer [9].

In humans, trophoblasts consist of several subpopulations [10–12]. The three primary classes of trophoblasts include cytotrophoblasts (CTBs), extravillous trophoblasts (EVTs), and the multinucleated syncytiotrophoblast (STB). CTBs represent the mononuclear, proliferative, undifferentiated trophoblast population of the placenta [10] whereas EVTs are highly invasive, immunosuppressive cells, that primarily serve to remodel the maternal spiral arteries [13–15]. Conversely, the STB is a terminally differentiated multinuclear trophoblast population that covers the entire surface of the chorionic villi. They serve dual roles as both a barrier to infection and a critical site layer for nutrient and gas transfer [16,17].

Unlike human trophoblasts, which have been heavily characterized and subtyped [12,18], the properties and subpopulations of swine trophoblasts are comparatively unknown, but several lines of evidence suggest that subpopulations exist. First, the point of contact between the fetal trophoblasts and uterine epithelium, termed the maternal–fetal interface, is extensively folded to enhance placental surface area [19]. Along these folds, it has been reported that aquaporins [20] and various transporters vary in expression [21], suggesting that trophoblast polarization along these folds occurs. Second, the porcine placenta contains areolae, a cluster of specialized, columnar trophoblast cells that sit atop uterine glands and facilitate the transfer of uterine gland secretions to the developing fetus [22]. It is known that areolar trophoblasts are primed to uptake uteroferrin compared to other trophoblasts [23] and express specific markers, such as cathepsin L1 (CTSL) [24], suggesting a unique pattern of differentiation. Lastly, recent studies in early gestation porcine placentas have revealed subpopulations of trophoblasts, which display unique expressional and putative functions [25,26]. While these data serve as a critical advancement, these studies are unable to spatially resolve these populations to known structures such as the interface or the areola, thereby limiting the interpretation of these data [25,26].

Another major hurdle to furthering our understanding of porcine placentation is the lack of suitable in vitro models. Limited access, coupled with high expense render in vivo and *ex vivo* models challenging. While porcine trophoblast cell lines, such as PTr2 [27,28], Jag-1 [29], and pTR cells [30] are available, these cells likely represent pre-implantation trophoblasts. Whether these cell lines recapitulate later stages of placental development, or the cellular heterogeneity observed in vivo remains unclear.

In human and murine systems, organoids have proven to be a valuable tool for studying complex biological systems in part because they can be cryopreserved, are scalable, and are often tractable [31]. Furthermore, as organoids spontaneously differentiate into various cell populations, they represent one of the most physiologically relevant in vitro models [31]. We and others have successfully derived trophoblast organoids from human and non-human primate (NHP) placentas, demonstrating that they recapitulate the cellular heterogeneity and gene expression patterns of the human and NHP placenta [32–37]. Because of the success of trophoblast organoid models, we sought to develop a similar model of the swine placenta. Here, we show the derivation of swine trophoblast organoids (sTOs) from full-term placental tissue. Our data demonstrate that sTO can be expanded, passaged, and cryopreserved, and recapitulate the expression of canonical markers of the swine placenta. Using Visium spatial transcriptomics of mid-gestation swine placentas, we identified the gene signatures of diverse trophoblast populations in vivo and identified novel maternal–fetal crosstalk pathways. By coupling spatial transcriptomics of whole placentas with single-cell transcriptomics from sTO, we found that organoids spontaneously differentiated into various trophoblast populations, accurately replicating their gene signatures. Additionally, we identified gene changes along a differentiation trajectory in sTOs, revealing signatures associated with the differentiation of distinct swine trophoblast subpopulations. Taken together, these data demonstrate that sTO provide an accurate and comprehensive model of the swine placenta, offering a powerful tool for advancing studies of porcine placentation and reproductive biology in vitro.

## Results

### Derivation and long-term expansion of swine trophoblast organoids

To derive sTOs, we collected full-term swine placenta tissue and applied a similar two-step enzymatic digestion protocol used previously for the derivation of human TOs (hTOs) [33] (Fig 1A). Isolated stem/progenitor cells were resuspended in Matrigel and allowed to self-organize over a period of 7–10 days until spherical structures, reminiscent of hTOs, were observed via brightfield microscopy (Fig 1B and 1C). Organoids were isolated from initial wells by picking out single organoids, dissociated them, and then passaging using a combination of enzymatic and mechanical dissociation. sTO growth was robust and scalable, with an average propagation interval of 5–7 days. Unlike hTOs, which develop with "mini-cavities" at their centers [32], sTOs exhibited a denser structure with no visible central cavities, which may reflect species-specific differences in tissue architecture and cellular organization (Fig 1C and 1D). Similar to hTOs, sTOs could be continuously passaged with no apparent phenotypic changes (e.g., organoid size, morphology, growth kinetics) and can be cryopreserved and revived. We successfully derived sTOs from five unique swine placentas with 100% success across all tissues collected.

Having successfully derived sTO, we then sought to confirm their epithelial origin. To do this, we performed immunofluorescence staining for various known markers of the porcine maternal–fetal interface. Notably, sTOs exhibited strong expression of the proliferation marker MKI67 (Fig 1E), with heterogeneous staining intensity across cells. This pattern suggests a transition from a highly proliferative state toward differentiation. Similarly, we found that sTO were positive for the epithelial marker cytokeratin 18 (Fig 1F) and the tight junction marker ZO-1 (Fig 1G), both of which are known to be expressed by porcine trophoblasts. These data suggest that sTO contain a highly proliferative population of epithelial cells derived from the maternal–fetal interface.

A limitation of hTOs is their inverted polarity when grown in Matrigel "domes", resulting in an inward-facing STB that does not accurately mimic the cellular polarity of the human placenta in vivo [32]. To define the cellular orientation of

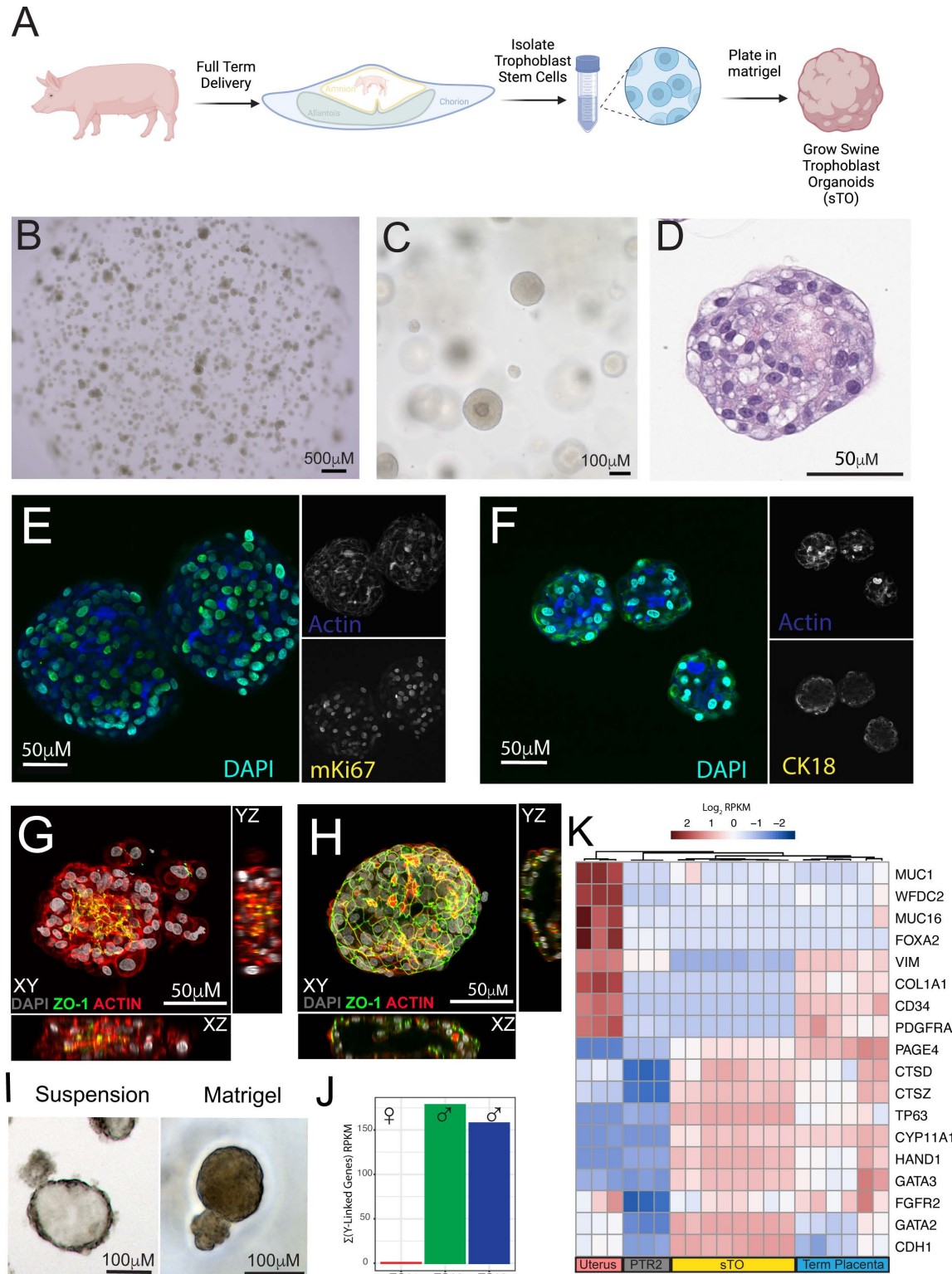

**Fig 1. Establishment and validation of swine trophoblast organoids (sTO). A)** Schematic of sTO derivation workflow. Created in BioRender. McCutcheon, C. (2025) https://BioRender.com/ohwh6rx. **B)** Representative low magnification brightfield image from a single well of sTO. **C)** High magnification brightfield image of sTO grown in Matrigel. **D)** Representative Hematoxylin and Eosin staining of a sectioned sTOs. **E, F)** Immunofluorescence

microscopy image of sTO grown in Matrigel stained with E) mKi67 and F) cytokeratin 18. **G)** Immunofluorescence microscopy image of sTO grown in Matrigel "domes". Apical marker ZO-1 is in green, DAPI in gray, and Actin in red. **H)** Immunofluorescence microscopy image of sTOs grown in suspension for 48 hours. Apical marker ZO-1 is in green, DAPI in gray, and Actin in red. **I)** Representative brightfield image of sTO grown in Matrigel "dome" (right) and suspension (left). **J)** Bar-plot showing the average summation of RPKM for nine Y-linked genes across 3 individual sTO lines. Black dots indicate replicates. sTO lines are denoted on the x-axis. **K)** Heatmap of known pig placentae and uterus markers expressed across Uterus, PTr2 cells, sTOs, and term pig placentae. Individual columns indicate replicates. Clustering of columns is performed using hierarchical clustering, which groups columns with similar expressional patterns. The underlying data for this figure can be found at the Gene Expression Omnibus via accession numbers GSM8980984, GSM8980985, GSM8980986, GSM8980987, GSM8980988, GSM8980989, GSM8980990, GSM8980991, GSM9083242, and GSM9083243.

sTOs, we performed immunostaining with the tight junction marker ZO-1, which localizes to the apical surface of swine trophoblasts in vivo [38]. We found that sTOs expressed ZO-1, which localized to the inner surface of organoids when grown in Matrigel (Fig 1G). We showed previously that hTOs cultured in suspension grow with more physiologic polarity, with the STB on the outer surface [32]. Using a similar approach, we found that sTOs also reversed their polarity when grown in suspension for approximately 48 h and that ZO-1 relocalized to the organoid outer surface (Fig 1H). sTOs grown in suspension also displayed a more cystic morphology (Fig 1I). This suggests that the polarity of sTOs is easily reversed, allowing for flexibility in experimental design.

To assess the ability of sTOs to recapitulate tissue-specific gene expression profiles, we performed bulk RNA sequencing on sTOs. Using a previously established protocol for porcine sex determination which relies on the expression of key Y-linked genes, we determined that sTO were derived from both male and female placentae (Fig 1J) [39]. Having confirmed that sTO were indeed fetal derived, we then compared their expressional profile to publicly available expressional datasets of full-term porcine placental tissue, porcine uterine tissue, and PTr2 cells, a commonly used in vitro model of swine trophoblasts. This approach allowed us to compare the global transcriptional landscapes of organoids and their corresponding tissues, identifying key expressional patterns to indicate how closely sTOs mimic the native tissue profile. This analysis revealed that sTOs also express high levels of canonical swine trophoblast markers, at similar levels to porcine placental tissue., whereas PTr2 cells exhibited very low levels of the same markers (Fig 1K). Conversely, we found that while uterine tissues expressed high levels of known uterine gland and fibroblast markers, sTO displayed an absence of expression of these markers. While term placentas did exhibit some expression of fibroblast markers, they similarly expressed an absence of uterine gland markers (Fig 1K). Together, these data indicate that sTO are composed of fetal-derived swine trophoblasts and display higher expression of these markers than conventional models such as PTr2 cells.

### Single-cell transcriptional profiling reveals cellular heterogeneity in sTOs

To determine whether sTOs differentiated into multiple trophoblast lineages, we next performed single-cell RNA-sequencing on three unique lines of sTOs. This analysis revealed five cell clusters (Fig 2A), with similar proportions across lines (Fig 2B). Canonical trophoblast markers, including KRT7, KRT18, KRT8, CDH1, GATA2, HAND1, TP63, and GATA3, were highly expressed in sTOs (Fig 2C) with low expression of the fibroblast markers VIM, PDGFRA, and COL1A1 (Fig 2D). Similarly, we observed very low expression of the uterine gland markers FOXA2, MUC16, MUC1, and CD36 (Fig 2D). Taken together, these data indicate that sTO are a pure population of trophoblasts that do not contain fibroblasts or glandular epithelial cells. Cluster gene expression enrichment analysis revealed that trophoblast clusters were discernable by cluster-specific gene expression (Fig 2E and S1 Table). For example, cluster 0 was enriched with transcripts such as SOX5 and WNT6, cluster 1 had enriched expression of tight junction-associated genes OCLN and CLDN1, cluster 2 was marked by cell proliferation genes like TOP2A, cluster 3 showed high expression of inflammatory genes including CXCL8 and interferon-stimulated genes (ISGs), and cluster 4 exhibited elevated levels of COX enzymes (Fig 2E). Previous studies using in situ hybridization, antibody staining, and single-cell transcriptomics have identified markers of both swine

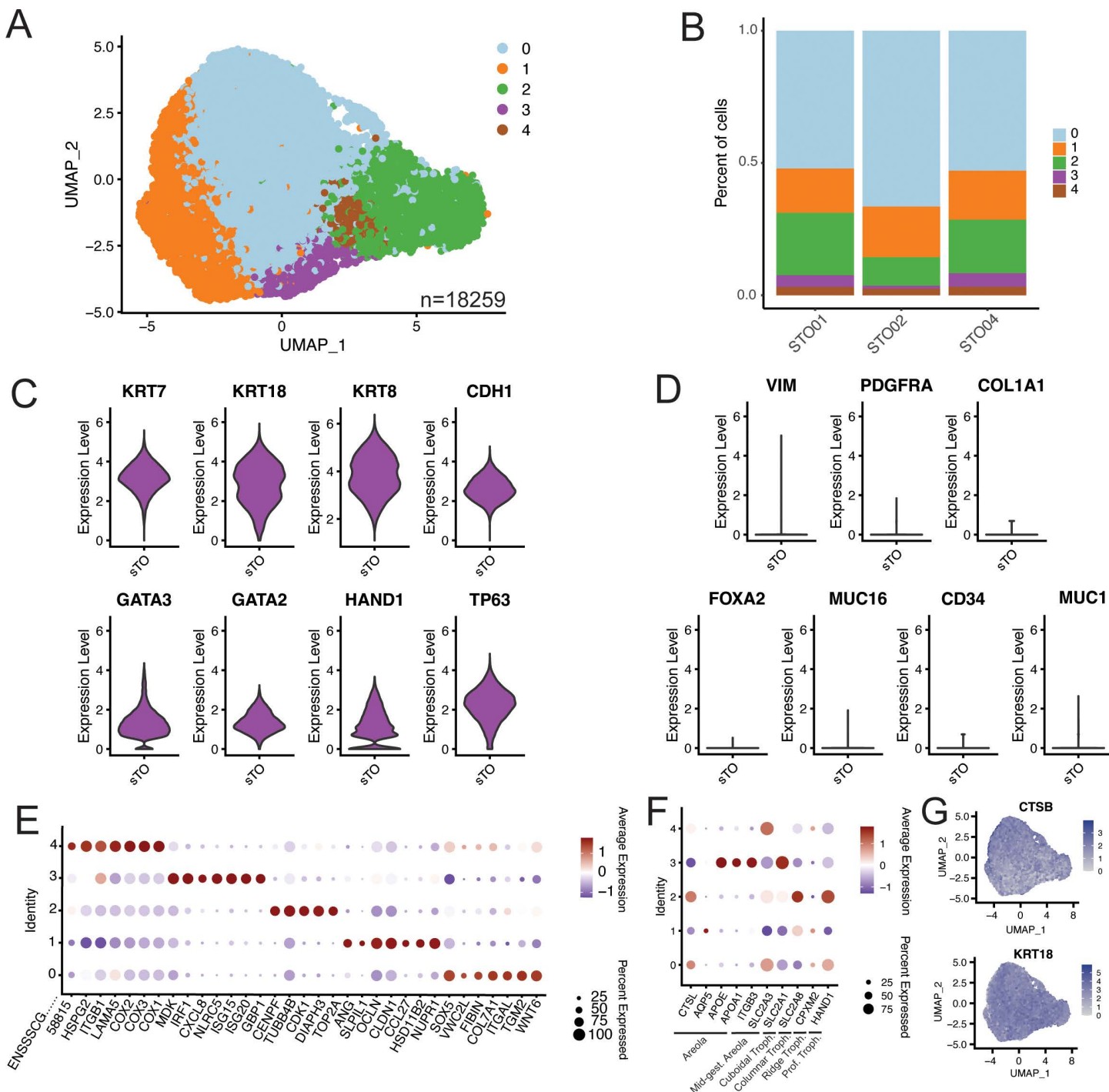

**Fig 2. Single-cell transcriptomics of sTOs reveals cellular heterogeneity. A)** UMAP from single-cell transcriptomics of 3 independent lines of sTOs. **B)** Stacked bar-plot showing the relative proportions of various cell populations across three sTO lines. **C)** Violin plot showing the expression of canonical trophoblast markers in sTO single-cell data. **D)** Violin plot showing the expression of canonical fibroblast markers in sTO single-cell data. **E)** Dot-plot showing the top enrichment markers for various cell populations in sTOs. Color of dots denotes average expression, whereas size represents percentage of cells in a cluster expressing the gene of interest. **F)** Dot-plot of canonical markers of various porcine trophoblast populations. Color of dots denotes average expression, whereas size represents percentage of cells in a cluster expressing the gene of interest. **G)** Feature plot showing wide-spread expression of pan-trophoblast markers in sTOs. The underlying data for this figure can be found at the Gene Expression Omnibus via accession numbers GSM8980992, GSM8980993, and GSM8980994.

interface trophoblasts and areolar trophoblasts [20,24,25,40,41]. While we observed expression of these markers within sTO cell populations, they did not fully segregate the trophoblast populations, ultimately limiting the utility of these markers to define the cellular identity of sTO-derived clusters (Fig 2F). Of note, we did observe that known pan-trophoblast markers such as CTSB and KRT18 were uniformly expressed in sTOs suggesting that these various sTO cell populations are distinctive lineages rather than contaminating cell populations (Fig 2G). While this analysis identified several trophoblast populations in sTOs, the lack of a comprehensive marker list specific to the swine maternal–fetal interface made it difficult to definitively assign cellular identities to these clusters.

## Visium-based spatial transcriptomics reveals cellular architecture at the swine maternal–fetal interface

To define the transcriptional signature of various cell populations comprising the porcine maternal–fetal interface, we utilized Visium spatial transcriptomics, which allows for high-resolution spatial mapping of gene expression. We applied this approach to mid-gestation maternal–fetal interfaces (uterus with attached placenta) from four separate pregnancies (2 male and 2 female), enabling the precise localization of gene expression patterns within the tissue architecture (Fig 3A). Our analysis identified 10 distinct cell populations among the four placentae analyzed (Figs 3B and S1A). To assign cellular and regional identities to these clusters, we utilized a combination of known marker expression (Fig 3C), histology (Fig 3D), and cluster-specific gene enrichment analysis (S2 Table). Notably, the maternal and fetal compartments were also discernable using this clustering technique (Figs 3E and S1B). The maternal compartment was divided into six clusters: the myometrium, represented by clusters 0 (Myo-1) and 6 (Myo-2); the endometrium, represented by cluster 1 (Endo); blood and blood vessels, represented by clusters 7 (Blood) and 4 (Vasc); and the uterine epithelium, represented by cluster 3 (UtEp) (Figs 3D, 3F and S1C). The fetal compartment was divided into four cell populations: the interface trophoblasts, represented by cluster 5 (IntTroph); the areolae, represented by clusters 8 (Areola-1) and 9 (Areola-2); and the interstitium, represented by cluster 2 (Ist) (Figs 3D, 3F and S1C). We observed some degree of sample-to-sample variability, which stemmed primarily from differences in both cellular makeup and tissue structure between sections (Fig 3G). Myometrium clusters were primarily defined by the expression of smooth muscle markers, including SMTN, CNN1, ACTG2, MYL9, and MYLK. Blood vessels were characterized by the expression of CLDN11 and CCL21, while blood cells were classified by high levels of ALAS2 and HBM (Fig 3C). The endometrium was marked by elevated expression of SLPI and MEP1B, both well-known to be expressed in this tissue (Fig 3C). Uterine epithelial cells expressed high levels of established markers IGFBP2 and IGFBP3, whereas interface trophoblasts showed elevated expression of pregnancy-stimulated genes (PSGs), PAGE4, PAG2L4, and HSD3B1 (Fig 3C). The interstitium cluster was defined by the expression of APOE, PEG10, DLK1, and GPC3 (Fig 3C). Finally, the areola clusters were divided into two histologically and transcriptionally distinct populations: Areola-1, characterized by high CTSL expression, and Areola-2, with moderate CTSL expression. Areola-1 also expressed high levels of LIF and MMP25, while Areola-2 was distinguished by elevated expression of LRP2, CYP11A1, and MMP7 (Fig 3C and 3H). Where Areola-2 displayed a structure that is continuous with the interface trophoblasts, Areola-1 did not appear to be continuous with the interface trophoblasts (Fig 3D, 3H and S1C), suggesting these distinct structures may have different functions.

## Defining cellular markers at the swine maternal–fetal interface through spatial transcriptomics

Using the spatial transcriptomics dataset described above, we analyzed the cellular and molecular landscape of the swine maternal–fetal interface to identify distinct markers associated with specific cell types and regions. This approach enabled us to map the spatial distribution of key genes, including those enriched in select trophoblast populations. To identify trophoblast specific markers, we performed differential expression analysis between clusters containing trophoblasts, (clusters 5, 8, and 9) and all other clusters. This analysis revealed both pan-trophoblast markers, and markers of specific trophoblast populations (Figs 4A, 4B and S2A–S2E). Specifically, the genes encoding cathepsin Z (CTSZ) and cathepsin B (CTSB) were highly expressed in both interface trophoblasts as well as areolae trophoblasts, but only weakly expressed

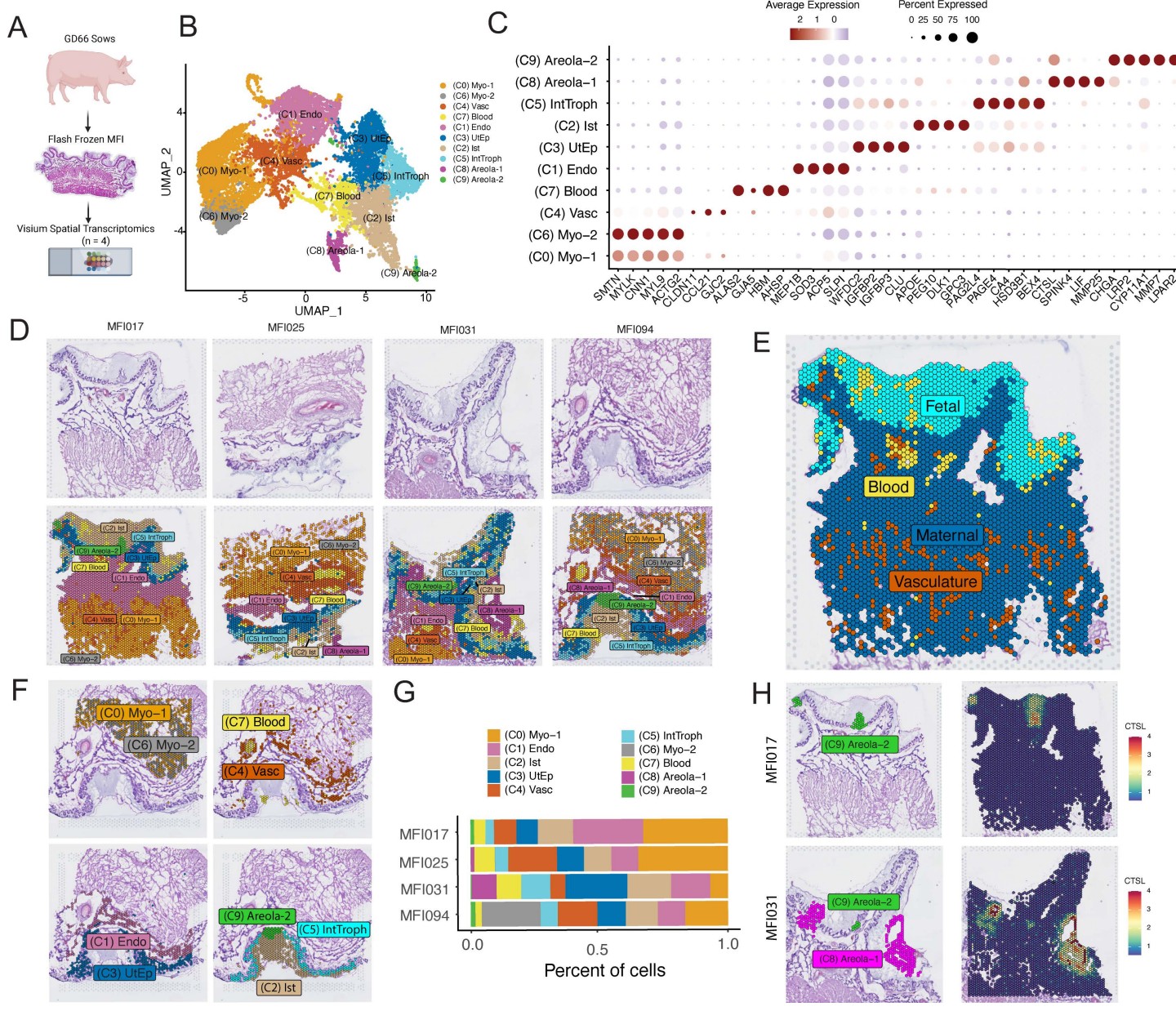

**Fig 3. Spatial transcriptomics resolves global expression at the porcine maternal–fetal interface. A)** Schematic of spatial transcriptomics workflow of gestational day 66 (GD66) maternal–fetal interfaces (*n* = 4; 2 male and 2 female). Created in BioRender. Mccutcheon, C. (2025) https://BioRender.com/hus6ktj **B)** UMAP showing 10 distinct populations obtained via spatial transcriptomics. **C)** Dot-plot of top markers for each cluster obtained via spatial transcriptomics. Color of dots denotes average expression, whereas size represents percentage of cells in a cluster expressing the gene of interest. **D)** H&E staining of individual porcine maternal–fetal interface sections (top). Spatial Dim-Plot showing the histologic localization of UMAP cluster populations (bottom). Each image represents a separate maternal–fetal interface. **E)** Spatial Dim-Plot showing separation of maternal and fetal compartments. Representative maternal–fetal Interface shown. **F)** Spatial Dim-Plot showing the histologic localization of UMAP individual cluster populations. Representative maternal–fetal Interface shown (MFI094). **G)** Stacked bar-plot showing the relative proportion of each cluster across individual maternal–fetal interfaces. **H)** Spatial Dim-Plot (Left panels) and Spatial Feature plots (Right panels) showing the localization of areola-1 and areola-2 structures (Left panels) and the expression of CTSL (right panels) within representative maternal–fetal interfaces. The underlying data for this figure can be found at the Gene Expression Omnibus via accession numbers GSM8980980, GSM8980981, GSM8980982, and GSM8980983.

 

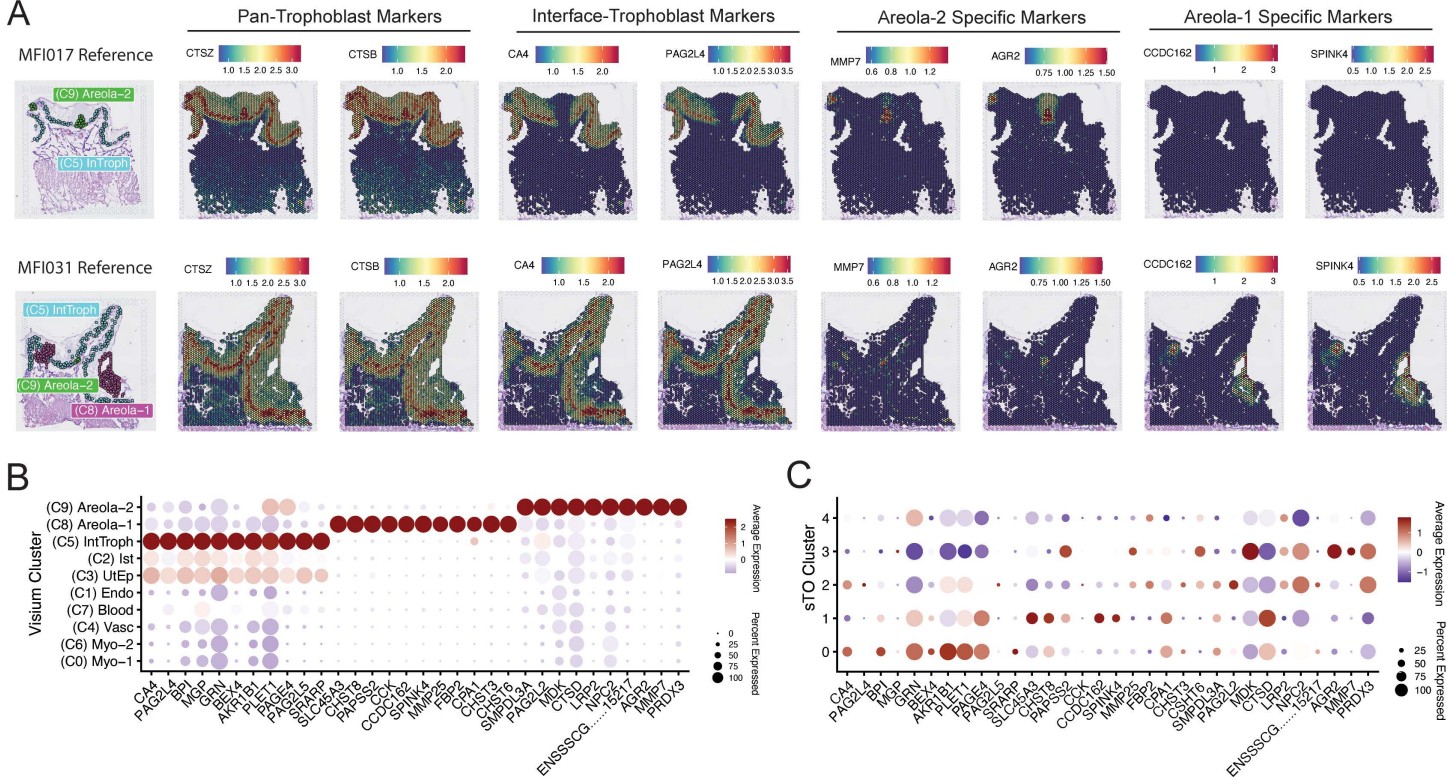

**Fig 4. Spatial transcriptomics identifies novel trophoblast specific markers. A)** Spatial Feature Plots showing the localization of known or novel pan-trophoblast markers, interface-trophoblast markers, areola-2 and areola-1 specific markers. Two representative maternal–fetal interfaces are shown. The reference localization of these structures are shown on the far-left panels. **B)** Dot Plot showing expanded markers of various trophoblast populations within the maternal–fetal interface. Color of dots denotes average expression, whereas size represents percentage of cells in a cluster expressing the gene of interest. **C)** Dot Plot showing the cluster-specific expression of known and novel trophoblast markers from (B) in sTOs. Color of dots denotes average expression, whereas size represents percentage of cells in a cluster expressing the gene of interest. The underlying data for this figure can be found at the Gene Expression Omnibus via accession numbers GSM8980980, GSM8980981, GSM8980982, and GSM8980983.

elsewhere (Figs 4A, 4B and S2A). Similarly, the genes encoding carbonic anhydrase 4 (CA4) and pregnancy-associated glycoprotein L4 (PAG2L4) demarcated interface trophoblasts (Figs 4A, 4B and S2B). Areolar trophoblasts universally expressed CTSL (Figs 3H and S2C); however, areola-1 trophoblasts specifically expressed coiled coil domain containing 162 (CCDC162) and serine peptidase inhibitor kazal type 4 (SPINK4) (Figs 4A, 4B and S2E). Conversely, areola-2 trophoblasts were marked by an upregulation of matrix metalloprotease 7 (MMP7) and anterior gradient protein 2 homolog (AGR2) (Figs 4A, 4B and S2D). Notably, all of these highly specific markers were also detected within sTO, with varied expression levels across clusters, suggesting some degree of differentiation across sTO cell populations (Fig 4C). Although these markers partially resolved the clusters, their leaky expression across multiple clusters limited our ability to accurately assign cellular identities.

## Characterizing cell–cell interactions within the porcine maternal–fetal interface using spatial transcriptomics

Maternal–fetal crosstalk has been widely studied in human pregnancy, where it plays crucial roles in nutrient exchange, immunological tolerance, and successful placentation [42–44]. Recent evidence suggests that similar communication processes take place in the peri-implantation porcine embryo and the maternal endometrium, highlighting the importance of defining these interactions in the swine placenta [26,45]. To delineate communication networks at the swine maternal–fetal

interface, we used CellChat to define cell-cell signaling pathways. CellChat is based on a comprehensive dataset of known ligand-receptor pairs, using their expression to infer communication between cells. In spatially resolved data, CellChat also incorporates the physical distance between tissue sites, filtering out interactions that are unlikely to occur due to the spatial separation between these sites. Using this approach we found that the myometrium (Myo, which was merged into a single cluster for this analysis) and the vasculature (Vasc) were responsible for the most robust communication, while the interstitium (Ist), endometrium (Endo), Areola (merged into a single cluster), and IntTroph showed the weakest number of communication networks (Fig 5A and 5B). Notably, all maternal clusters were able to crosstalk with the fetal trophoblast populations except the myometrium and vasculature. Using CellChat, we then classified both outgoing and incoming signaling patterns. For outgoing signaling, pattern 1 and pattern 2 were largely derived from the maternal compartment, with pattern 1 including the endometrium and blood, whereas pattern 2 is primarily derived from the myometrium and vasculature. Conversely, pattern 3 was shared among fetal-derived populations, including the Ist, IntTroph, and Areola (Fig 5C). We similarly observed that incoming patterns were similarly segregated by fetal and maternal compartments, with pattern 1 exhibiting the strongest signaling for Areola, Ist, and IntTroph, and pattern 2 being composed of Myometrium, Vasculature, and Blood (Fig 5C). Each pattern for the outgoing and incoming signaling networks corresponded to distinct communication networks. For example, outgoing pattern 1 was dominated by OSTN, VEGF, and BMP signaling, whereas pattern 3 was dominated by HGF, EGF, and CDH signaling. Similar observations were made for incoming signaling patterns, with pattern 1 corresponding to high levels of EGF, ANGPT, and GRN signaling, and pattern 2 being characterized by high levels of NPR1, KIT, and CXCL signaling (Fig 5C). Notably, this compartmentalization of the maternal and fetal signaling suggests that there is extensive maternal–fetal crosstalk occurring in the porcine placenta. Some of these signals were unidirectional, with all outgoing signals deriving from the maternal compartment, such as the HSPG signaling pathway, which is largely derived from the myometrium and vasculature. Interestingly, this HSPG2 signal, which is important for angiogenesis, is capable of being sensed by both the maternal system, and the interface trophoblasts, largely due to the restriction of the receptor DAG1. In contrast, many crosstalk pathways were enriched in various placenta trophoblast subpopulations. For example, the MK pathway was restricted primarily to areola trophoblasts, whereas the IL6 pathway was largely restricted to the interface trophoblasts. In both cases, the outgoing signal, MDK and IL6, were localized to the areola and interface trophoblasts, respectively. While the MDK receptor SDC2 was broadly expressed in both the maternal and fetal compartment, IL6R was largely restricted to the endometrium, with some lower-level expression in all other tissues. Interestingly, in both the MK and IL6 pathways we observed some degree of inter-trophoblast communication, with interface trophoblast and areolar trophoblasts displaying some degree of crosstalk. Together, these findings highlight the extensive, bidirectional maternal–fetal crosstalk occurring at the porcine maternal–fetal interface.

## Cell type deconvolution of trophoblast populations in sTOs using spatial transcriptomics

To define trophoblast populations within sTOs, we performed cell type deconvolution by integrating the single-cell transcriptomics dataset from sTOs with the spatially resolved transcriptome of the swine maternal–fetal interface. To do this, we filtered the Visium dataset to only include trophoblast populations, specifically clusters 5, 8, and 9 (Fig 6A–6C). Next, we integrated the single-cell RNA-sequencing data from sTO with the Visium spatial transcriptomics dataset. For cell type prediction, we applied a previously established probabilistic modeling algorithm, which leverages the likelihood of gene expression patterns across datasets to assign cluster identities in a spatial context. This analysis enabled us to predict the identities of sTO clusters with high confidence. We found that sTO cluster 0 very strongly mapped to interface trophoblasts, whereas sTO clusters 1, 3, and 4 map to areola-2 and areola-1 trophoblast clusters (Figs 6D, 6E, S3A and S3B). In contrast, cluster 2 did not exhibit strong mapping to any structure (Figs 6D, 6E, S3A and S3B). However, this cluster expressed high levels of the proliferation-associated genes TOP2A and MKI67, suggesting that it represented the stem/progenitor cell population, which is likely present at very low abundance in tissue at the stage of gestation when we were collecting the tissues (Fig 6F).

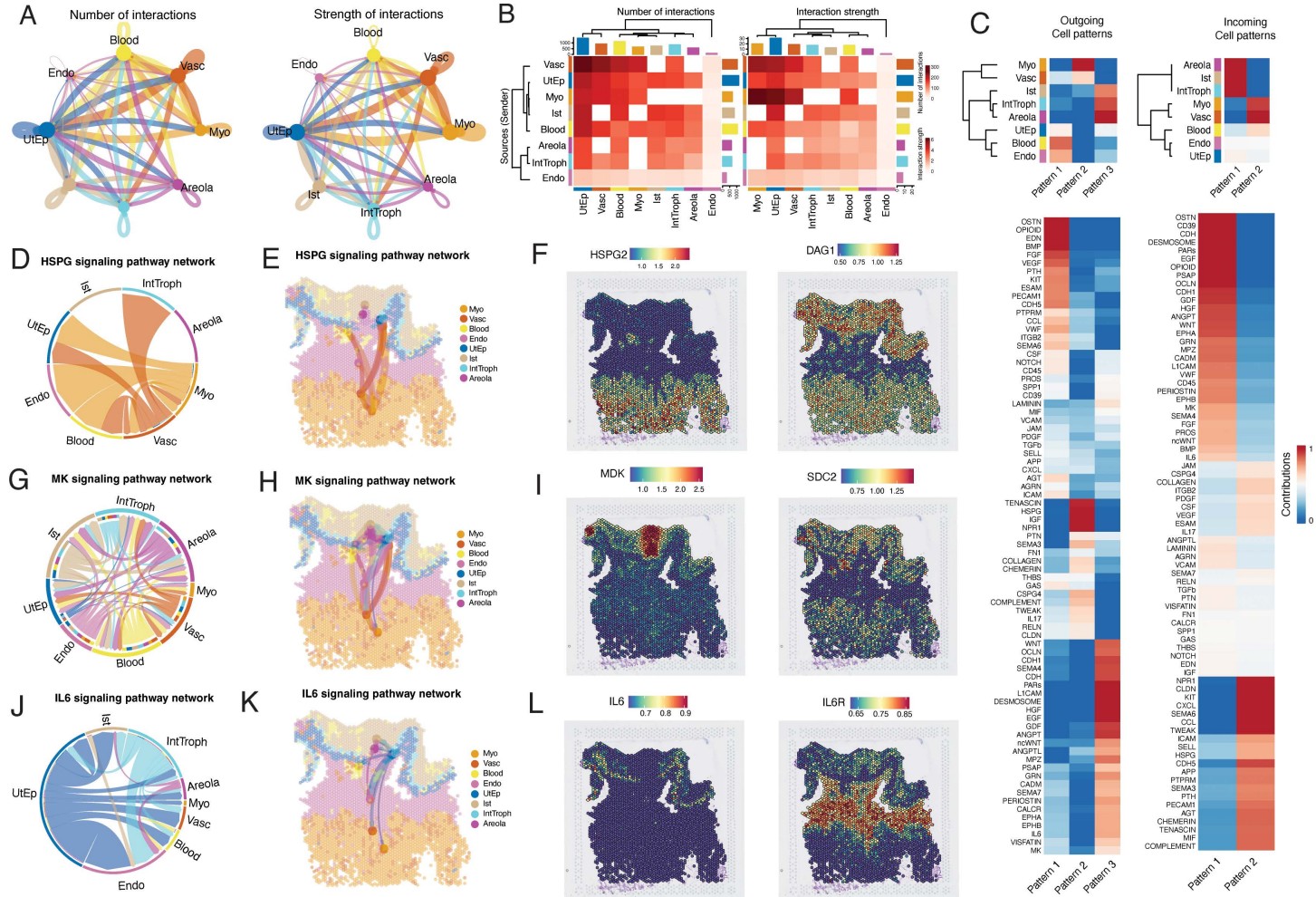

**Fig 5. Cell–cell interactions within the porcine placenta maternal–fetal interface. A)** Chord diagrams showing the number (Left panel) and strength (right panel) of calculated inter-cluster interactions within the porcine maternal–fetal interface. **B)** Heatmap showing the number (left panel) and strength (right panel) of interactions between clusters within the porcine maternal–fetal interface. Color indicates increased number (left) or strength (right) between the sending cluster and the receiving cluster. **C)** Heatmaps showing the outgoing (upper left panel) and Incoming (upper right panel) signaling patterns by cluster for Visium spatial transcriptomics. Coloring indicates relative cluster contribution to a pattern with red being high and blue being low contribution level. Lower panels similarly indicate the contribution a specific signaling pathway to the corresponding pattern designation with the lower right panel indicating outgoing signaling patterns and the lower left panel indicating incoming signaling patterns. **D, E)** Plots showing HSPG signaling within the porcine maternal–fetal interface, with **D)** being a chord diagram and **E)** showing the spatial localization of these interactions. **F)** Spatial feature plot showing the localization of various genes associated with the HSPG pathway. The right panel shows the localization of HSPG2, whereas the left panel shows the localization of DAG1. **G, H)** Plots showing MK signaling within the porcine maternal–fetal interface, with **G)** being a chord diagram and **H)** showing the spatial localization of these interactions. **I)** Spatial feature plot showing the localization of various genes associated with the MK pathway. The right panel shows the localization of MDK, whereas the left panel shows the localization of SDC2. **J, K)** Plots showing IL6 signaling within the porcine maternal–fetal interface, with J) being a chord diagram and **K)** showing the spatial localization of these interactions. **L)** Spatial feature plot showing the localization of various genes associated with the IL6 pathway. The right panel shows the localization of IL6, whereas the left panel shows the localization of IL6R. The underlying data for this figure can be found at the Gene Expression Omnibus via accession numbers GSM8980980, GSM8980981, GSM8980982, and GSM8980983.

We then identified shared markers between sTO clusters and their corresponding spatial transcriptomics clusters to determine the gene expression signatures that defined these mappings. We found that sTO cluster 0 cells expressed 59 markers that were conserved in interface-trophoblasts, which included AKR1B1, HAVCR1, C1QTNF4, and TGM2

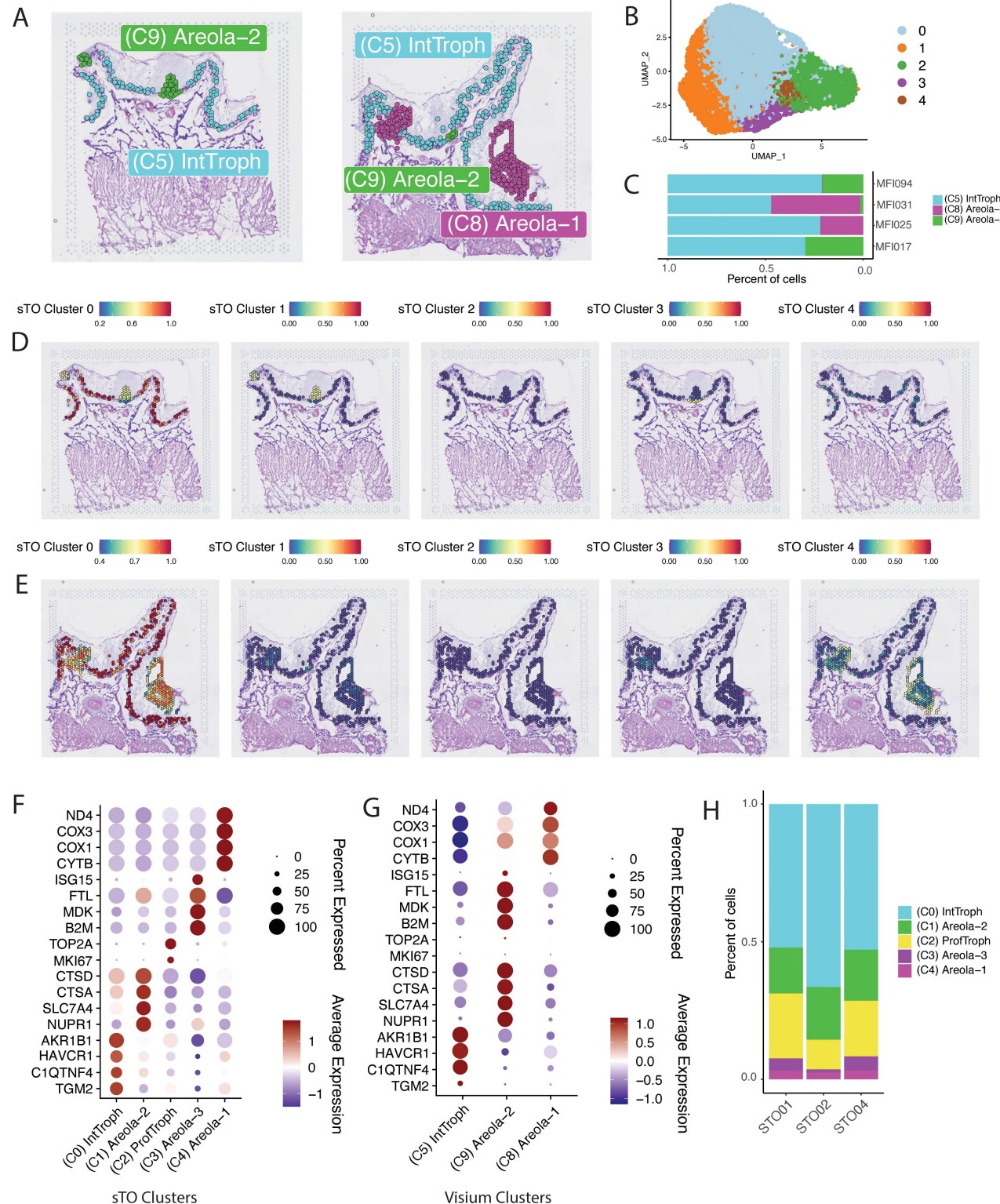

**Fig 6. Integration of sTOs and spatial transcriptomics datasets enables precise cellular identification and classification. A)** Spatial Dim-Plot showing facetted clusters from two representative maternal–fetal interfaces. As shown Areola-1, Areola-2, and Interface-Trophoblasts were included in this analysis. **B)** UMAP of sTOs clusters obtained via single-cell RNA-sequencing (*n* = 3 lines). **C)** Stacked bar-plot showing relative proportions of

various clusters across individual maternal–fetal interfaces. **D, E)** Spatial feature plot showing prediction scores for each spot in the spatial dataset subsetted by sTO cluster. Color shown indicates probability that the classified sTO cluster is localized to a given position within the spatial dataset. **F, G)** Dot-plot showing the localization of shared markers between the sTO clusters **(F)** and maternal–fetal interface clusters **(G)** Color of dots denotes average expression, whereas size represents percentage of cells in a cluster expressing the gene of interest. **H)** Stacked Bar Plot showing the relative proportion and identity of sTO cell populations. The underlying data for this figure can be found at the Gene Expression Omnibus via accession numbers GSM8980980, GSM8980981, GSM8980982, GSM8980983, GSM8980992, GSM8980993, GSM8980994.

(Fig 6F and 6G and S3 Table). Similarly, sTO cluster 1 expressed 63 shared markers with areola-2 cluster, including cathepsins A and D (CTSA and CTSD), SLC7A4 and NUPR1 (Fig 6F and 6G and S4 Table). sTO cluster 3 had 110 shared markers with areola-2 cluster, with both clusters highly expressing ISG15, FTL, MDK, and B2M (Fig 6F and 6G and S5 Table). Conversely, sTO cluster 4 expressed only 20 shared markers to areola-1 cluster, largely including COX1, COX3, ND4, and CYTB (Fig 6F and 6G and S6 Table). Using this approach, we were able to assign cellular identities and identify shared gene expression profiles for all trophoblast populations within sTOs, which were distributed at similar ratios across independent lines (Fig 6H). Taken together these data indicate that sTO are able to spontaneously differentiate into various populations of trophoblasts found at the porcine maternal–fetal interface.

## Pathway analyses reveals conserved functional niches of sTO cell populations

In vivo, interface trophoblasts and areola trophoblasts have distinct functions. Our data indicated that sTOs differentiated into both of these populations, yet whether these populations retained the functional capacity of their in vivo counterparts remained unclear. To this end, we examined the functional pathways active in both sTOs and tissues across trophoblast subpopulations. Given that IntTroph and Areola-2 clusters had the highest probability scores and shared markers between sTO and tissue, we chose to focus this analysis on these two clusters. We performed differential expression analysis between interface trophoblasts and areola-2 trophoblasts, finding that expressional patterns between these two trophoblast populations were starkly different in both the tissue and sTO (S4A and S4B Fig). Using GO-Term analysis, we identified enriched pathways for areola-2 and interface trophoblasts, several of which were enriched in both sTOs and tissue analysis (S7–S10 Tables). Notably, we found that interface trophoblasts are enriched in GO terms such as extracellular matrix, focal adhesions, and angiogenesis (S7 and S9 Tables). Conversely, areola-2 clusters were enriched primarily in GO terms surrounding endosome formation (S8 and S10 Tables). Expression of various endosome-related genes demonstrated that areola-2 clusters in sTOs and tissue Visium datasets are highly enriched in endosomal markers, whereas interface trophoblasts distinctly lack endosomal gene expression (S4C and S4D Fig). Taken together these data demonstrate that sTOs retain at least some key functions of the corresponding tissue populations, highlighting their utility for studying swine placental biology.

## Trajectory inference reveals trophoblast differentiation pathways in sTOs

Organoids provide a powerful model for defining differentiation trajectories, as they enable the study of cell fate transitions from progenitor cells to more differentiated cell states in a controlled environment. We applied Slingshot trajectory analysis to sTOs to trace the dynamic progression of trophoblast cells along distinct developmental pathways. To do this, we first subsetted the sTOs single-cell dataset to include major cell populations, which included proliferating trophoblasts (ProfTroph), interface trophoblasts (IntTroph), and Areola-2 trophoblasts. We then performed Slingshot trajectory analysis by first setting the starting point at the proliferating trophoblasts, which displayed the highest expression of known stemness markers, including MKI67 and TOP2A. We then allowed inference of downstream differentiation paths without further constraint. This analysis revealed a single differentiation trajectory, starting in ProfTroph, progressing through IntTroph, and terminating in Areola-2 (Fig 7A). To define gene expression changes along this trajectory, we applied the fitGAM function

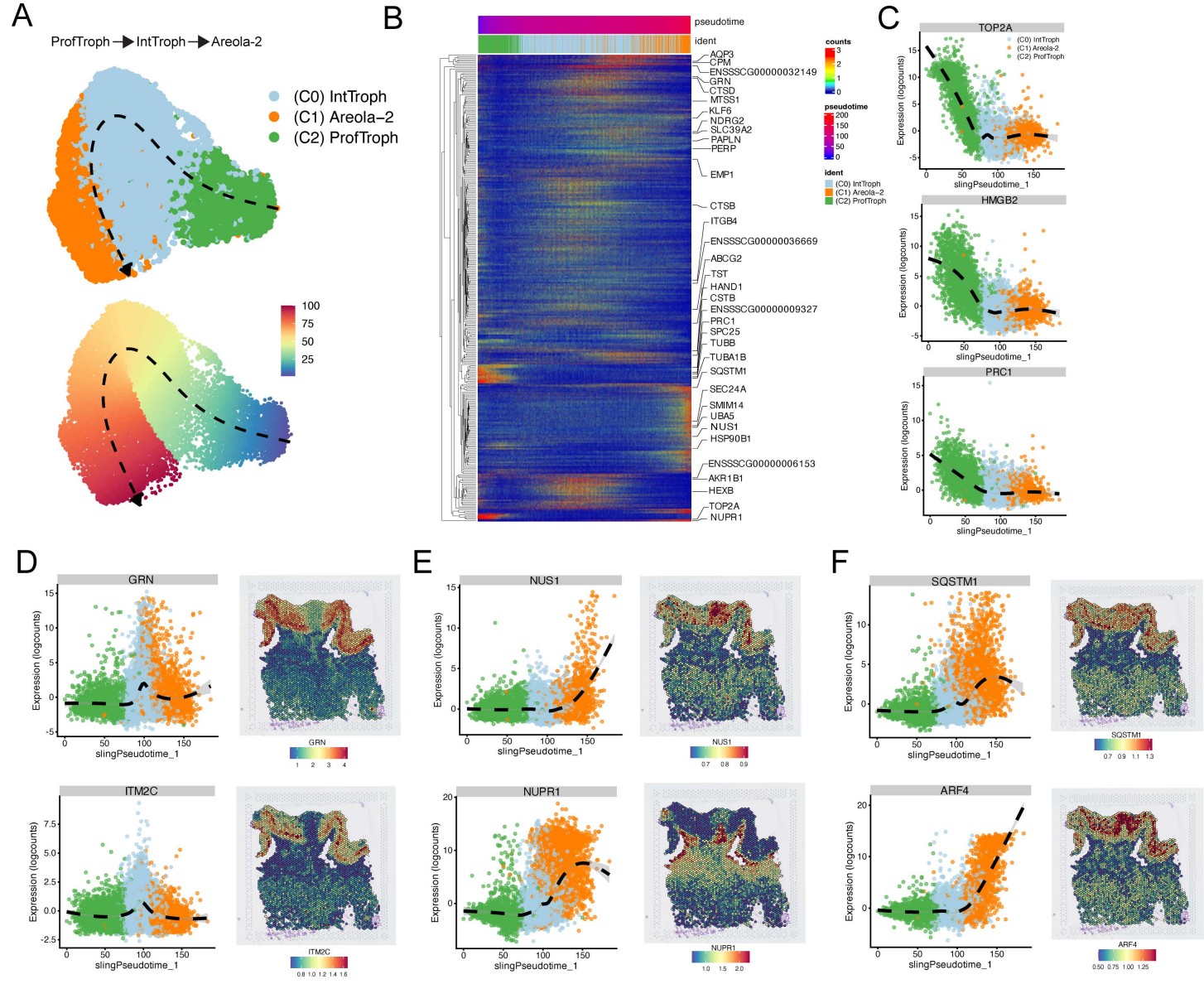

**Fig 7. Trajectory analysis reveals the differentiation pathway of swine trophoblasts. A)** UMAPs showing Slingshot calculated trajectory of trophoblasts within sTO. Top panel shows the trajectory across labeled clusters, whereas bottom panel shows the trajectory across a pseudotime. **B)** Heatmap showing temporally regulated genes across the calculated pseudotime trajectory. Individual columns represent individual cells that are ordered along the calculated pseudotime. Identities are positions along the pseudotime are shown above the heatmap. **C)** Representative expression plots showing gene expression changes along the calculated pseudotime that are enriched in the proliferative stem cell population. **D)** Representative expression plots and spatial feature plots showing genes that are enriched during the transition to Interface-Trophoblasts and their expression pattern within our spatial transcriptomics dataset. **E, F)** Representative expression plots and spatial feature plots showing genes that are enriched during the transition to Areola-2 trophoblasts and their expression pattern within our spatial transcriptomics dataset. The underlying data for this figure can be found at the Gene Expression Omnibus via accession numbers GSM8980980, GSM8980981, GSM8980982, GSM8980983, GSM8980992, GSM8980993, GSM8980994.

to model dynamic gene expression patterns to assess how specific gene expression profiles evolved across the different stages of trophoblast differentiation. Many genes were temporally regulated along the trajectory, with some showing distinct patterns of upregulation and downregulation, while others exhibited cyclical regulation across the different stages

of trophoblast differentiation (Figs 7B and S5 and S11 Table). Notably, TOP2A, HMGB2, and PRC1 were strongly down-regulated along the trajectory, suggesting a loss of a proliferative capacity as differentiation progresses (Fig 7B and 7C). In contrast, GRN and ITM2C, both of which are known to be expressed in the swine placenta, were markedly increased in expression in IntTroph, but then became downregulated in the Areola-2 differentiated state (Fig 7D, left). Similarly, several genes, including NUS1, NUPRR1, SQSTM1, and ARF4, all increased in expression at the terminal areola-2 differentiation stage (Fig 7E and 7F, left). To validate these findings from the trajectory analysis of sTOs, we compared the gene expression patterns to the Visium spatial transcriptomics data. This comparison revealed a high level of concordance between the temporally regulated markers identified in sTOs and their expression in corresponding tissue structures, further confirming that sTOs effectively recapitulate the differentiation processes observed in vivo (Figs 7D–7F and S5A–S5F).

## Comparative cell–cell interaction network analysis identifies preserved communication networks within sTOs and the swine placenta

Crosstalk within the human placenta is well documented and controls processes such as fetal development and immunological tolerance. Our data indicates that the porcine placenta similarly displays a high level of intraplacental crosstalk, but additionally displays crosstalk between various trophoblast subpopulations, including crosstalk between the areolar trophoblasts and interface trophoblasts. Given that sTO contain both areolar and interface trophoblast populations, we wanted to better characterize these inter-trophoblast crosstalk events within the porcine placenta and assess how well these interactions are recapitulated in sTOs. To do this, we first subsetted both the Visium data and sTO datasets to only contain areolar trophoblast and interface trophoblast subpopulations. For clarity, areola trophoblast clusters were combined into a singular cluster for each dataset. We then used Cellchat to assess cellular communication pathways present between these populations. This analysis revealed extensive crosstalk in sTOs and in tissue derived trophoblasts with substantial overlap in the signaling pathways between these systems (Fig 8A and 8B). Of note we found 28 shared signaling pathways between sTO and tissue trophoblasts (Fig 8A). In both sTO and tissue trophoblasts, we observed that interface trophoblasts provide more substantial signaling than areolar trophoblasts (Fig 8B).

Patterning analysis revealed that the senders and receivers, were conserved across several of these signaling pathways, suggesting concordance between sTO and tissue trophoblast signaling (Fig 8C–8F). Some of these pathways displayed bidirectional crosstalk between interface trophoblasts and areolar trophoblasts, such as the ANGPTL and CDH pathways, which displayed extensive autocrine and paracrine signaling (Fig 8G–8N). The ANGPTL pathway displayed a high degree of both autocrine and paracrine signaling between interface and areola trophoblasts, primarily due to the widespread expression of integrins, syndecans, and ANGPTLs (Fig 8G–8J). While we similarly found that areolar trophoblasts display both autocrine and paracrine signaling in the CDH pathway, we found that interface trophoblasts do not display the same ability to signal in an autocrine manner, likely due to the lack of CDH4 expression (Fig 8K–8N). In contrast, we found that some pathways were cell type specific, such as the AGRN pathway, which is restricted to interface trophoblasts (Fig 8O and 8P). While we detected high levels of DAG1 in both populations, we found that AGRN was restricted largely to interface trophoblasts (Fig 8Q and 8R). Importantly, we found that in many of these pathways, including AGRN, CDH, and ANGPTL, displayed a high degree of similarity in signaling patterns between these systems highlighting the utility of sTO in studying inter-trophoblast communication in the placenta (Fig 8G–8R).

## Discussion

US agricultural commodities from swine account for approximately $62 billion in added economic value and is heavily reliant on successful animal husbandry [46]. In all eutherian mammals, including swine, successful placentation is critical for fetal growth and survival [7]. Therefore, understanding non-human placental function, development, and defenses is critical for ensuring continued reproductive success. While there is growing interest in studying non-human placentation, the lack of appropriate models remains a major hurdle. We and others have demonstrated that trophoblast organoids derived

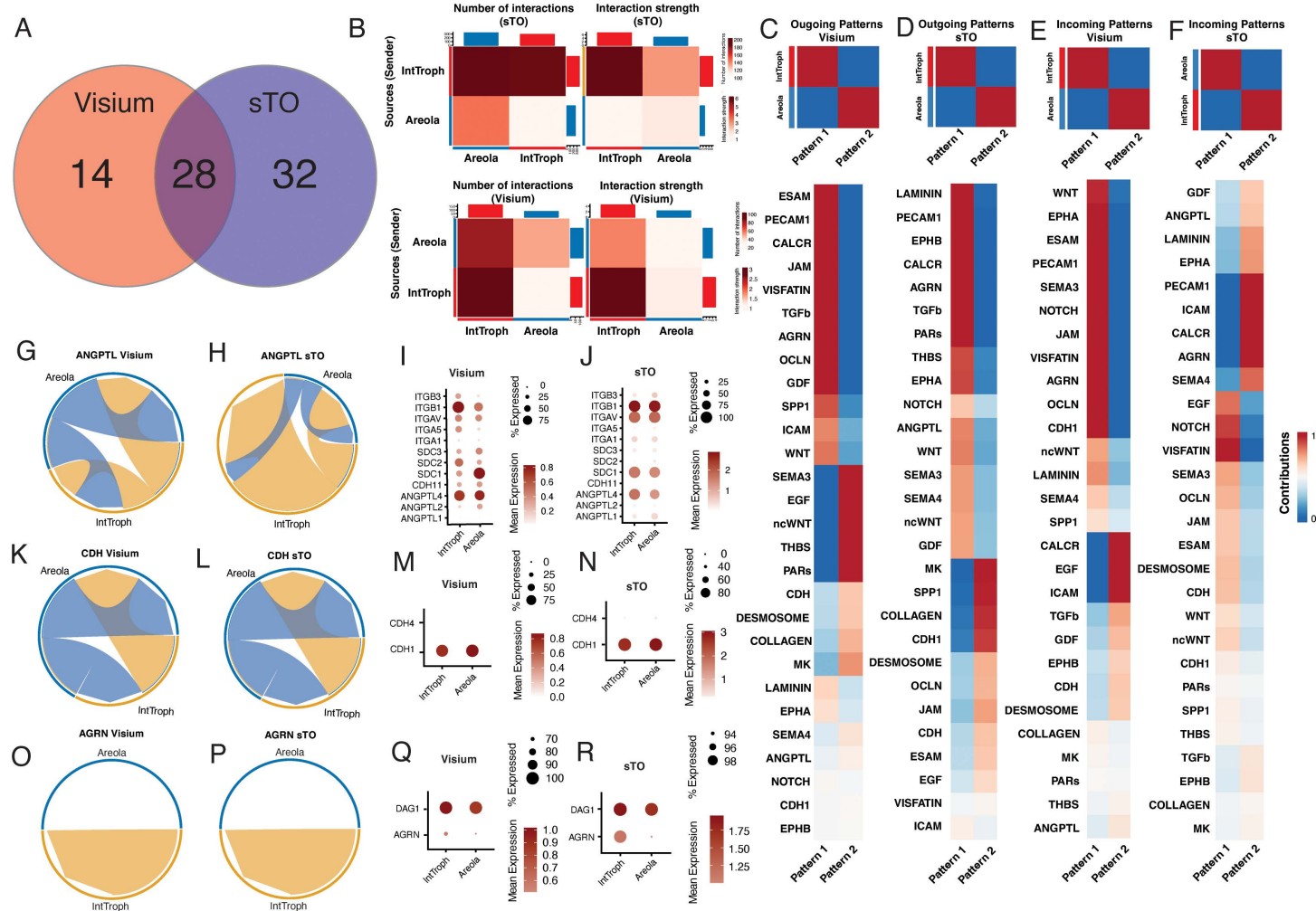

**Fig 8. sTOs recapitulate the trophoblast crosstalk pathways observed in the porcine placenta. A)** Venn Diagram showing the number of unique and shared trophoblast communication pathways between sTO and porcine trophoblasts in vivo. **B)** Heatmaps showing the number (top left panel) and strength (top right panel) of interactions between clusters within sTO. Similarly, the bottom panels show the number (bottom left panel) and strength (bottom right panel) of interactions within porcine trophoblasts from spatial transcriptomics data. In both cases Color indicates increased number (left) or strength (right) between the sending cluster and the receiving cluster. **C–F)** Heatmaps showing the signaling patterns (top panels) and the shared signaling pathways contributing to those patterns, for sTO and porcine trophoblasts from spatial transcriptomics. **C, D)** shows the outgoing patterns for **C)** porcine trophoblasts obtained via spatial transcriptomics and **D)** sTO. **E, F)** shows the incoming patterns for **E)** porcine trophoblasts obtained via spatial transcriptomics and **F)** sTO. For upper panels, coloring indicates relative cluster contribution to a pattern with red being high and blue being low contribution level. For lower panels, color indicates the relative contribution of a pathway to a given pattern. **G, H)** Chord diagrams showing ANGPTL signaling within both **G)** Visium trophoblasts and **H)** sTO. **I, J)** Dot plots showing the expression level of pathways within the ANGPTL pathway for **I)** Visium trophoblasts and **J)** sTO. Color of dots denotes average expression, whereas size represents percentage of cells in a cluster expressing the gene of interest. **K, L)** Chord diagrams showing CDH signaling within both **K)** Visium trophoblasts and **L)** sTO. **M, N)** Dot plots showing the expression level of pathways within the CDH pathway for M) Visium trophoblasts and N) sTO. Color of dots denotes average expression, whereas size represents percentage of cells in a cluster expressing the gene of interest. **O, P)** Chord diagrams showing AGRN signaling within both **O)** Visium trophoblasts and **P)** sTO. **Q, R)** Dot plots showing the expression level of pathways within the AGRN pathway for **Q)** Visium trophoblasts and **R)** sTO. Color of dots denotes average expression, whereas size represents percentage of cells in a cluster expressing the gene of interest. The underlying data for this figure can be found at the Gene Expression Omnibus via accession numbers GSM8980980, GSM8980981, GSM8980982, GSM8980983, GSM8980992, GSM8980993, GSM8980994.

from human placentae are a critical tool for the study of human placentation due to their ability to be passaged, cryopreserved, and to recapitulate human trophoblast heterogeneity [32,34,47]. Here, we report the establishment of sTOs, which recapitulate key aspects of the swine placenta.

We show that sTOs can be derived from full-term swine placentae, enabling model establishment without the need to sacrifice animals. Like hTOs, sTO can be expanded, cryopreserved, and successfully recovered, making them an ideal in vitro model. Current models, such as PTr2 cells, contain a unique expressional profile to swine placental tissue, and express low levels of several trophoblast markers. We show here that sTOs express high levels of the trophoblast markers KRT18, ELF3, and GATA3, and an expressional signature more consistent with swine placentae. Importantly, sTOs cultured within Matrigel spontaneously form an interiorly localized apical surface that is reversible by culturing in suspension, thereby allowing for customization of this model. Together, our data suggest that sTOs better recapitulate the transcriptional signature of swine placental tissue than currently available in vitro models.

As discussed previously, the swine placenta is distinct from the human placenta at both a structural and cellular level [7]. In humans, placental trophoblasts fall into several distinct subpopulations, including CTBs, EVTs, and the STB, all of which have distinct roles in the formation of the hemochorial placenta [7]. In the epitheliochorial placenta, such as those found in swine, there is no direct contact between maternal blood and fetal trophoblasts, which theoretically could limit the efficiency and molecular breadth of nutrient transfer to the developing fetus [7,8]. To facilitate this exchange, the highly folded interface expresses various nutrient transporters and aquaporins [20,21]. Further, the porcine placenta contains a specialized structure known as an areola, which is a dome like cluster of trophoblasts that sit atop uterine glands [22]. These structures, which have a peak density of approximately 2,500 per placenta [48], collect and transfer maternally-derived histotroph to the fetus as a nutrient source. Indeed, previous studies have indicated that areola surface area is directly correlated with fetal growth, further supporting their critical role in nutrient acquisition [48]. Interestingly, areola structures have been noted in many ungulates, including hippopotamuses [49], dolphins [50], and llamas [51], suggesting that areola are critical for fetal growth across species.

Several studies have indicated that these porcine trophoblast subpopulations have some degree of cellular heterogeneity in vivo. It is known that along placental folds of the interface, expression of various transporters varies. For example, GLUT1 and GLUT8 have opposing expression profiles, with GLUT1 being located at the base of the folds and GLUT8 being localized to the tip of the folds [40]. Similar spatially localized gene expression has been observed for aquaporins [20]. Additionally, trophoblasts localized to the areola have increased expression of CTSL, AQP5, and ITGB3 relative to other trophoblast populations [20,24,41]. Furthermore, how these various populations differentiate and develop is largely unknown, primarily due to the lack of an in vitro model that provides this cellular heterogeneity without the need to consistently access fresh tissue.

We and others have previously demonstrated that in human trophoblast organoids, there is extensive cellular heterogeneity, indicating that trophoblast organoids spontaneously differentiate into the various trophoblast populations of the placenta [32–34,52]. Single-cell RNA-sequencing of sTO revealed similar cellular heterogeneity within this organoid model. While previous studies have identified markers of various trophoblast subpopulations in swine placentae [20,24,40], these markers were insufficient to segregate sTO cell populations. Using Visium spatial transcriptomics of mid-gestation swine maternal–fetal interfaces, we defined the transcriptomic landscape of swine trophoblast populations in situ. By coupling a single-cell RNASeq dataset from sTOs and Visium spatial transcriptomics dataset from swine placentae, we were able to assign identities to sTOs populations using an unbiased probabilistic modeling approach. Simultaneously, we identified a shared gene expression profile between sTO cell populations and their corresponding placental cell populations. Using this approach, we demonstrate that sTOs contain a proliferative stem cell population, an interface trophoblast population, and three areola trophoblast populations. Critically, this approach allowed for robust identification of cell populations in an unbiased manner.

Using trajectory analysis, we defined how sTOs gain their heterogeneity. Using Slingshot, we demonstrate that proliferative stem cells differentiate into interface trophoblasts, which can then differentiate into areola-2 trophoblasts. This analysis uncovered numerous expression changes throughout the differentiation process, including novel targets like ITM2C and NUPR1, which could be investigated in future mechanistic studies to enhance our understanding of swine placental development. Having defined the expressional changes along this trajectory, we then confirmed their localization in tissue finding high concordance between sTOs and tissue-level expression, suggesting a shared trajectory between these two systems. By comparing global gene expression changes in sTO-derived interface trophoblasts and areola-2 trophoblasts, we go on to show that these two populations display starkly different functional niches, with interface trophoblasts being primed to form focal adhesions and promote angiogenesis, whereas areola-2 trophoblasts were enriched in programs for the formation of endosomes. It is noted within the literature that areola trophoblasts contain a high number of intracellular vesicles, presumably to facilitate transport of histotroph across this epithelial barrier [22]. As expected, when we examined the expression of key endosome markers within the tissue and sTO, we observed similar patterns of expression, with endosome markers expressed at a higher level in areola-2 cells compared to interface trophoblasts. Together, these data demonstrate that sTOs recapitulate the developmental and functional differences in trophoblast populations observed in the porcine placenta, highlighting their utility in studying porcine placental development.

While combining sTO and spatial datasets allowed for robust cell cluster identification, parallel analyses enabled us to further explore the functional similarities between sTOs and placental tissue. Using CellChat, we first identified dozens of signaling pathways occurring at the porcine maternal–fetal interface. Interestingly, we found several instances of inter-trophoblast signaling within the porcine placenta. By applying CellChat to the shared trophoblast populations found within our sTO and placental tissue datasets, we revealed several key signaling pathways are conserved between these systems. For example, we that the angiopoietin-like proteins (ANGPTL), Agrin (AGRN), and Cadherin (CDH) pathways are highly conserved within sTO and tissue trophoblasts. These pathways have been shown to be active in the placenta and play important roles in regulating epithelial to mesenchymal transitions, basement membrane function, and nutrient metabolism [53–56]. Importantly, sTO recapitulated autocrine and paracrine signaling patterns observed in placental tissue, highlighting their utility as a physiologic model. Importantly, these analyses were based on known human receptor-ligand interactions and rely on shared annotations between the human and porcine genome. It remains possible that some receptor–ligand pairs in the porcine genome are not annotated, which would preclude them from being included as an interaction in our results. Therefore, this represents a conservative estimate of cell-cell communication in the porcine placenta. We believe that these data provide a robust framework for hypothesis generation for understanding signaling within the porcine placenta.

Previous studies have used single-cell RNA-sequencing to characterize the porcine uterus and embryos [25,26,45,57]. However, these studies rely on existing markers or GO-term-based functions to assign cellular identity, lack histological structural context, do not provide spatial segregation of gene expression signatures, and were not performed at a gestational stage where critical structures, such as the areola, are present. Our data robustly define the expression profiles of various histological structures at the swine maternal–fetal interface. By mapping gene expressions directly to corresponding histological structures, we confirmed cellular identities without relying on conventional markers, making this assignment unbiased. Moreover, using mid-gestation swine maternal–fetal interfaces, we were able to characterize the expression profiles of key structures, including the areola and the interface, which have previously eluded detailed analyses. This dataset provides new insights into placental biology that were not achievable with earlier approaches.

Demonstrating the utility of this spatially resolved dataset, we identified two transcriptionally and structurally distinct areola populations within the porcine placenta: Areola-2, which appears continuous with the interface trophoblasts, and Areola-1, which is localized primarily to the placental interstitium without evidence of direct contact with the interface. This finding emphasizes the power of spatial transcriptomics in revealing previously unrecognized cellular diversity within

structurally complex tissues such as the placenta. While both clusters expressed high levels of the known areola marker CTSL [24], these clusters had divergent gene expression profiles with Areola-2 expressing high levels of MDK, CTSD, PAG2L2, and Areola-1 expressing high levels of CCDC162, SPINK4, and MMP25. While it remains unclear why these two areola populations exist, previous results [24] have shown that histologically similar CTSL positive structures exist within the porcine placenta. It is possible that these represent areola at different stages of development, with Areola-1 representing a more mature population due to its higher expression of CTSL and its increased size. Together, these findings highlight the power and utility of the spatial transcriptomic dataset in advancing our understanding of the fundamental biology of the porcine placenta.

Our work offers a comprehensive characterization of spatial gene expression in the porcine placenta and successfully establishes an organoid model that captures its cellular heterogeneity. However, limitations include the resolution of Visium spatial transcriptomics, which does not allow for single-cell level definition of histological populations. For example, the identified Visium clusters may represent a mixture of gene expression profiles but nonetheless captures the dominant signals of key structures and cell populations. For instance, although we observed some overlap in marker expression between uterine epithelium and trophoblast populations, these clusters still separate along the interface. Moreover, the populations are strongly represented by known markers, indicating that despite resolution constraints, this technique effectively segregates distinct populations and provides robust biological insights. Another limitation of this study is that we focused on a single mid-gestation time point. While this provides valuable insights into the biology of the porcine placenta during this stage, it limits our ability to assess dynamic changes in cellular composition and signaling across different stages of gestation. While previous studies have highlighted that human trophoblast organoids represent first trimester tissue [33,35], our bulk RNA-seq data demonstrated that sTOs have similar expression to porcine placentae regardless of gestation. It remains possible that early or later gestation placentae would have mapped with higher probabilities to sTO populations enabling more robust identification. Future studies aimed at capturing multiple gestational timepoints in the porcine placenta are therefore warranted. While the data presented here demonstrate the utility and strength of spatial transcriptomics, they represent only a portion of the insights generated in this study. The thousands of spatially resolved genes and pathways identified provide a valuable resource for advancing porcine reproductive biology. To facilitate broader access, we have developed a user-friendly web-based app, enabling researchers without advanced computational resources to explore and analyze the data (see Data availability statement for further details)

We believe this study serves as a foundational study for studying the porcine placenta, via the presentation of a physiologic model and a spatially resolved roadmap of the maternal–fetal interface. These critical findings allow for future studies to be proposed. For example, future studies aimed at studying the function of the signaling pathways identified here may discover new avenues to reduce reproductive failure, including failure caused by vertical transmission of pathogens or defective placentation. Additionally, these data may inform studies aimed at identifying the triggers of areola development, which may now be possible in vitro. We hope that these data will spark new interest in studying porcine reproductive biology.

Taken together, our study describes the derivation of an organoid model that recapitulates the cellular heterogeneity and functionality of the swine placenta. Additionally, we provide a spatially resolved expression landscape of the swine maternal–fetal interface, creating a valuable resource for the study of non-human placentation. The ability to map gene expression directly to histological structures offers unprecedented insights into the cellular dynamics of the porcine placenta, setting a new benchmark for comparative placental biology research. These findings have the potential to significantly advance our understanding of placental biology across eutherians, offering a critical framework for studying key aspects of maternal–fetal communication and placental development. Moreover, the organoid model and spatial transcriptomic data will serve as foundational tools for exploring mechanisms of placental function and dysfunction, with broader implications for reproductive biology and veterinary medicine.

## Methods

### Animal husbandry and tissue collection

**Tissue collection for sTO derivation.** Placentae for sTOs were collected from crossbred sows consisting of Yorkshire, Large White, and Landrace breeds from the North Carolina State University (NCSU) Swine Educational Unit (SEU), which is a facility free from Porcine Reproductive and Respiratory syndrome virus (PRRSV), swine influenza A virus, and mycoplasma species. The average gestation length of sows at the SEU is 114.2 ± 1.3 days. At SEU older parity sows ($n = 2$; parity 5) that didn't farrow by 114 days were induced using 2 mL of Lutalyse (Dinoprost tromethamine injection, Zoetis) intramuscularly, whereas gilts ($n = 1$) were allowed to farrow naturally. Placentae collected from gilts or sows were placedin ziploc bags containing Dulbecco's Modified Eagle Medium (DMEM, Cytiva Hyclone cat. # SH30081.01) supplemented with 10% inactivated fetal bovine serum (FBS, Biowest #S1620 Heat Inactivated, USDA approved origin), 1,100 IU/mL penicillin, 1,100 mg/mL streptomycin (Cytiva Hyclone 100× solution, #SV30010), 2 mM L-glutamine (GIBCO 100× 200 mM, #A2916801), Amphotericin B 2.5 µg/ml (Gibco, Fungizone, 15290-018)). Placentae were collected under experimental procedures approved by the NCSU Institutional Animal Care and Use Committee (IACUC) ID# 22–122-B ($n = 2$, gilts, IM Lutalyse injection) or via normal farrowing activities performed in NCSU SEU ($n = 3$)) and conducted in accordance with Animal Welfare Act regulations and according to the Guide for the Care and Use of Laboratory Animals.

**Tissue collection for visium spatial transcriptomics.** Animal procedures carried out in compliance with Purdue University's animal care policies and approved by the Institutional Animal Care and Use Committee (IACUC Protocol #2104002130), and as previously described [58]. Further, all animal protocols were conducted in accordance with Animal Welfare Act regulations and according to the Guide for the Care and Use of Laboratory Animals. In short, a total of $n = 3$ commercial landrace/large white composite gilts, were selected from the Animal Sciences Research and Education Center (ASREC) at Purdue University, synchronized with oral progestogen (Altrenogest at 17.6 mg/day for 14 days). Following withdrawal, the gilts were heat checked twice daily with an intact boar and artificially inseminated 12 h after the first observed lordosis responses, and every 24 h thereafter until the end of estrus. Pregnancy was confirmed via transcutaneous ultrasound on gestation day 30–35. Gilts were fed a standard gestation diet comprised of corn, soybean meal and dried distillers grain, and formulated to meet the dietary requirements as outlined by the NCR. On gestation day 65 (relative to first artificial insemination) pregnant gilts were stunned with a penetrating captive bolt and euthanized via rapid exsanguination. The gravid uterus carefully extracted though a midline incision and linearized such that each fetus and its corresponding placenta could be isolated. Sections of the maternal–fetal interface were collected adjacent to the umbilical stump and a roughly 1 cm$^2$ section of intact placenta (maternal and fetal tissue) was excised, embedded in optimal cutting temperature (OCT) media (Tissue-Plus OCT Compound; Fisher Scientific), and slowly cooled to cryogenic temperature using dry ice.

### Derivation, culture, and passaging of sTO

sTOs were derived using a previously established method used for the derivation of human trophoblast organoids [33]. Briefly, placentae were extensively rinsed with Phosphate-Buffered Saline (PBS) and sequentially digested with 0.2% trypsin-250 (Alfa Aesar, J63993-09)/0.02% EDTA (Sigma-Aldrich, E9884-100G) and 1.0 mg/mL collagenase V (Sigma-Aldrich, C9263-100MG). Digested tissue was then disrupted by pipetting up and down vigorously ~10 times with 10 mL serological pipette (VWR, 89130-898; MSP, 62-1005-10) and pooled prior to washing with Advanced DMEM/F12 medium (Life Technologies 12634-010). Pelleted digests were then resuspended in ice-cold growth-factor reduced Matrigel (Corning 356231). Matrigel 'domes' (40 µL/well) were plated into 24-well tissue culture plates (Corning 3526), placed in a 37°C incubator to pre-polymerize for ~3 min, turned upside down to ensure equal distribution of the isolated cells in domes for another 10 min, then overlaid with 500 µL prewarmed term trophoblast organoid medium (tTOM).

tTOM consisted of Advanced DMEM/ F12 (Life Technologies, 12634-010) supplemented with 1× B27 (Life Technologies, 17504-044), 1× N2 (Life Technologies, 17502-048), 10% fetal bovine serum (FBS) (vol/vol, Cytiva HyClone, SH30070.03), 2 mM GlutaMAX supplement (Life Technologies, 35050-061), 100 μg/mL Primocin (InvivoGen, antpm-1), 1.25 mM N-Acetyl-L-cysteine (Sigma-Aldrich, A9165), 500 nM A83-01 (Tocris, 2939), 1.5 μM CHIR99021 (Tocris, 4423), 50 ng/mL recombinant human EGF (Gibco, PHG0314), 80 ng/mL recombinant human R-spondin 1 (R&D systems, 4645-RS-100), 100 ng/mL recombinant human FGF2 (Peprotech, 100-18C), 50 ng/mL recombinant human HGF (Peprotech, 100-39), 10 mM nicotinamide (Sigma-Aldrich, N0636-100G), 5 μM Y-27632 (Sigma-Aldrich, Y0503-1MG), and 2.5 μM prostaglandin E2 (PGE2, R&D systems, 22-961-0).

Cultures were maintained at 37°C in a humidified incubator with 5% $CO_2$. Medium was renewed every 2–3 days. Small trophoblast organoid spheroids became visible by approximately 7–10 days post isolation. Derived sTO were passaged every 5–7 days depending on their size and density. To passage, spent media was removed from sTO and fresh Advanced DMEM/F12 added to the well. Domes were disrupted using a pipet tip and pelleted via centrifugation. Pellets were then digested in prewarmed TrypLE Express (Life Technologies, 12605- 028) at 37°C for 5–7 min. Pelleted organoids were then further dissociated manually using a P200 pipette by vigorously pipetting approximately 200–300 times. Dissociated sTOs were pelleted and resuspended in fresh ice-cold Matrigel and replated as domes at the desired density for continuous culture. All experiments described in this paper were performed in sTOs under passage 15 to limit potential passage level effects.

For cryopreservation, sTOs were grown to high density and spent media was aspirated. CryoStor CS10 (Stem Cell Technologies; # 07930) was then added 1mL per 'dome' and domes disrupted using a p1000 pipette. Organoids were then transferred to a sterile cryovial and gradually frozen to −80°C using a CoolCell (Corning). After 24 h vials were transferred to vapor phase liquid nitrogen for long-term storage. To thaw cryopreserved organoids, cryovials were rapidly thawed (<2 min) in a 37°C water bath with gentle agitation. Freezing media and organoids were then transferred to a 15mL conical and resuspended 9 mL of prewarmed Advanced DMEM/F12. Organoids were then pelleted and resuspended in ice-cold Matrigel and seeded into 'domes' in 24-well plates. Fresh tTOM, including 5 μM Y-27632 (Sigma-Aldrich, Y0503-1MG) was added prior to incubation for 24–72 h at 37°C with 5% $CO_2$. sTO were then passaged as described above.

### Reversal of sTO polarity

To reverse organoid polarity, we utilized a previously established protocol we developed to invert hTOs [32]. Briefly, spent media was removed and 1mL of ice-cold cell recovery media (Corning Cat#) added to each well. Domes were gently disrupted using a P1000 and organoids transferred to a 15 mL conical. Organoids were incubated for 30 min on ice to dissolve Matrigel prior to being washed twice using 1× DPBS. Organoids were then resuspended in 500 mL of prewarmed TOM and plated into ultra-low binding 24-well plates. Organoids were incubated statically at 37°C with 5% $CO_2$ for 48 h prior to usage.

### Histology and immunofluorescence staining of sTOs

Mature organoids were fixed using 4% paraformaldehyde in PBS for 15 min and subsequently washed with PBS. Histology was performed by HistoWiz (histowiz.com) using a Standard Operating Procedure and fully automated workflow. Samples were processed, sectioned at 5μm. Whole slide scanning (40×) was performed on an Aperio AT2 (Leica Biosystems). For immunofluorescence staining, whole-mount staining of organoids was performed using previously described methods with some modifications. Briefly, organoids were removed from PBS and permeabilized for 10 min in a permeabilization buffer consisting of 0.5% Triton X-100 and 1% Bovine Serum Albumin (BSA) in PBS. Organoids were then pelleted and stained overnight using a ZO-1 antibody conjugated to AlexaFluor 488 (Santa Cruz) at a dilution of 1:50 in permeabilization buffer. The organoids were then pelleted and washed twice in PBS, prior to staining with Phalloidin conjugated to AlexaFluor 633 (Thermo) according to manufacturer's instructions. Organoids were then pelleted and washed twice

in PBS prior to mounting with Vectashield containing DAPI. Slides were then imaged using Olympus Fluoview FV3000 inverted confocal microscope and images processed using FIJI. Brightfield images of whole organoids were obtained from fresh, mature, unfixed wells using a Keyence BZ-X810 all-in-one fluorescence microscope

## Bulk RNA-sequencing of sTO and term placenta tissues

**RNA isolation, library prep, and sequencing.** Total RNA was isolated with the Sigma GenElute total mammalian RNA miniprep kit (Cat # RTN350-1KT) or Qiagen RNAeasy Mini Kit (Cat #74104) following the manufacturer's instruction and using supplementary Sigma DNase digestion. RNA quality and concentration were determined using a Nanodrop ND-1000 Spectrophotometer. For bulk RNA-seq analysis, RNA was isolated from organoids or term placental tissue as described above. Purified Total RNA was verified by Thermo scientific Nanodrop one. The libraries were prepared by the Duke Center for Genomic and Computational Biology (GCB) using the Tru-Seq stranded total RNA prep kit (Illumina). Sequencing was performed on the NovaSeq 500 by using 75-bp paired-end sequencing.

For PTr2 cells, raw reads from a previously published study were downloaded from the Sequence Read Archive (SRA): SRR16912824, SRR16912825, and SRR16912827. Similarly, raw reads for tissues were downloaded from SRA under the accession numbers SRR651716, SRR651719, SRR651720 for uterine tissues, and SRR6703674, SRR6703675, SRR6703676, SRR6703677 for four of the term placentas.

**Bulk RNA-seq analysis.** Raw reads were aligned to the swine genome (susScr11) using QIAGEN CLC Genomics (v20) and mapped data were exported and compiled. The packages tidyverse, magrittr, pheatmap, rio, factoextra, ggforce, tidyheatmap, and DESeq2 were utilized for analysis. All analyses were performed using package specified methods. Heatmaps were made using pheatmap and tidyheatmap, with row scaling and hierarchical clustering of both rows and columns. For sex determination analyses an established protocol for sexing pigs using bulk rna-sequencing data was utilized, with minor modifications [39]. Briefly, the summation of the following 9 Y-linked genes: EIF2S3Y, ZFY, KDM5D, EIF1AY, DDX3Y, LOC100625207, LOC110257894, LOC110255257, and LOC110255320. Summation scores below 2 are considered female, whereas scores above 2 are considered male.

## Single-cell RNA-sequencing of sTOs

sTO from three independent lines (STO01, STO02, and STO04) were grown to high density using the method above. Spent media was removed from the wells and fresh Advanced DMEM/F12 added to the well. Domes were disrupted using a pipet tip and pelleted via centrifugation. Pellets were then digested in prewarmed TrypLE Express (Life Technologies, 12605-028) at 37°C for 10–15 min. Pelleted, digested organoids were then further dissociated using a 23G needle aspirating and ejecting 3 times. For STO01 and STO04 a dead cell removal step was performed (Miltenyi Biotech catalog #: 130-090-101) per manufacturer's instructions. Disassociated sTO were then filtered through a 40 μm cell strainer, which was subsequently washed using 4 mL of Advanced DMEM/F12. Pelleted cells were then resuspended in fresh Advanced DMEM/F12 and placed on ice until viability counts were performed.

Libraries were prepared following standard 10× Genomics Single Cell Gene Expression 3′ protocols. Samples containing highly viable cells (80% viability or higher), were washed and filtered through 40 μm filter and resuspended in a 1× PBS/ 0.04% BSA buffer, at a concentration of 1,000 cells/μL. A Cellometer K2 instrument (Nexcelom – Lawrence, MA) was then used to determine the concentration of the single-cell suspension. Cells were then suspended in master mix containing reverse transcription (RT) reagents using a volume equivalent to the target capture number of 10,000 cells (plus approximately 60% overage to account for capture efficiency of the assay). This master mix was loaded onto the 10× microfluidics chip, together with gel beads carrying the Illumina TruSeq Read 1 sequencing primer, a 16 bp 10× barcode, a 12 bp unique molecular identifier (UMI) and a poly-dT primer, and oil for the emulsion reaction. The Chromium X instrument was then used to generate gel beads in emulsion (GEMS) for the RT and barcoding of single cells. After the RT reaction, the GEMs were broken, full-length cDNAs were cleaned with Silane Dynabeads, and then amplified via PCR.

Following purification using SPRI bead size selection the cDNAs were assayed on a 4200 TapeStation 4200 High Sensitivity D5000 ScreenTape (Agilent – Santa Clara, CA) for qualitative and quantitative analysis.

Enzymatic fragmentation and size selection were used to optimize the cDNA amplicon size for the sequencing library preparation in which Illumina P5 and P7 sequences (San Diego, CA), a sample index, and TruSeq read 2 primer sequence are added via end repair, A-tailing, adaptor ligation, and PCR. The final libraries contained P5 and P7 primers used in Illumina bridge amplification. Following additional purification libraries were assayed for quality, using a TapeStation HSD1000 ScreenTape, then quantitated and checked for successful adapter ligation with the KAPA Library Quantification Kit (Roche-Indianapolis, IN). Sequences were generated using paired end sequencing on an NovaSeq X Plus, at ~62k reads/cell.

**Analysis of single-cell data**

**QC and cluster analysis.** FASTQ files were uploaded to the 10× Cloud Analysis website (https://www.10xgenomics.com/products/cloud-analysis) and analyzed using cellranger v7.1.0 using Sus scrofa genome v11.1 downloaded from Ensembl (www.ensembl.org). Mapped files were downloaded through command line and filtered feature matrices utilized for secondary analyses in R using a Seurat-based pipeline [59]. Briefly, the packages Seurat v4.4.0, SeuratObject v5.0.1, and SeuratWrappers v0.2.0, devtools 2.4.5, dplyr v1.1.4, harmony v1.2.0, and patchwork v1.2.0, were used for analysis. Individual imported filtered feature matrices underwent filtering for to only include cells with nFeatures falling between 2,500 and 15,000, nCounts lower than 200,000, percent mitochondrial reads falling between 0.05 and 1, and percent ribosomal reads lower than 3. Matrices were then merged, and individually normalized using SCTransform, regressing percent mitochondrial reads, percent ribosomal reads, nFeatures, and nCounts. Integration of the data was then performed on the SCT normalized data using SelectIntegrationFeatures (nFeatures = 4,000), PrepSCTIntegration, FindIntegrationAnchors, and the IntegrateData commands. Dimensional reduction using RunPCA and RunUMAP was then performed using the top 30 dimensions, followed by cluster analysis (FindNeighbors and FindClusters) at a resolution of 0.2. This resolution was selected because it was able to clearly resolve clusters by markers, with minimal overlap in markers between clusters, and to optimize probabilistic mapping to tissue in downstream analyses. Prior to subsequent analyses, where indicated, the RNA assay was normalized and scaled (NormalizeData and ScaleData, and ALRA (RunALRA) mediated imputation of the data was performed as previously described. Marker identification was performed using the FindAllMarkers function using the ALRA assay, with a minimum fold change threshold of 0.25 and detected in a minimum of 0.25 of cells. For marker analyses, the default Seurat adjusted $p$-value of $p < 0.01$ (Bonferroni correction) was considered significant.

**Pseudotime analysis.** The differentiation trajectories from Seurat-identified clusters were determined using the Slingshot package (version 2.6.0) in R [60]. The root was set at the proliferative trophoblasts, which expressed the highest levels of stemness markers, and allowed inference of downstream differentiation paths without further constraint. The raw RNA count data and the generated Slingshot object were then utilized to run the evaluateK() function, testing a range of 3–9 knots. The optimal number of knots was determined to be 4. Subsequently, the fitGAM() function in tradeSeq (version 1.5.10) was applied using this optimal value, and lineage-specific gene expression was identified through the associationTest() function [61]. Heatmaps of expression changes along the lineages were created using ComplexHeatmap() on log-transformed counts, and were rasterized using the "Bessel" filter in ImageMagick [62]. Finally, the plotGenePseudotime() function was used to visualize raw count gene expression across individual cells along the lineages derived from the Slingshot object.

**Differential expression and GO-term analysis.** To perform differential expression between clusters, the FindMarkers function was utilized setting the LogFoldChange threshold to 0.01. Significant hits were defined as having a Log2FoldChange of at least 0.25 and an adjusted $p$-value < 0.01 (Bonferroni correction). GO-Term was performed using DAVID pathway analysis Significant hits were then input into the Database for Annotation, Visualization and Integrated

Discovery (DAVID; https://david.ncifcrf.gov/home.jsp) and GO-Terms exported. Significantly enriched GO-Terms were defined as GO-Terms with a *p*-value <0.05 after Bonferroni correction.

**CellChat analysis for sTOs.** Cellchat for sTO were performed using previously established methods [63,64]. Briefly, interaction inputs, complex inputs, cofactor inputs, and gene info inputs were downloaded from CellChatDB and cellchat objects were created using the ALRA assay. Overexpressed genes and interactions were then identified using the identifyOverExpressedGenes and identifyOverExpressedInteractions functions and gene expression projected onto a protein-protein network using projectData. Communication probabilities were then calculated using the computeCommunProb function. Communications were then filtered to include interactions including a minimum of 10 cells (filtercommunication function) and probabilities calculated using computeCommunProbPathway function. Aggregated networks were then calculated using aggregateNet and centrality calculated using netAnalysis_computeCentrality function. Patterning analysis was performed first by selecting a number of patterns using the selectK function, which was assigned based on when the which Cophenetic and Silhouette values droped suddenly. Pattern plots were generated using the identifyCommunicationPatterns function. Additional plots were generated using standard methods as described in detailed scripts.

## Spatial transcriptomics of mid-gestation pig placentae

Tissues embedded in OCT compound were sectioned at 10 μm thickness and placed on a Tissue Optimization (TO) Slide to determine optimal permeabilization conditions. After establishing the conditions, tissue sections were positioned within a 6.5 mm$^2$ capture area on an expression slide containing 5,000 spatially barcoded probes. The tissues were subsequently fixed and stained with Hematoxylin and Eosin. Following staining, the tissues were permeabilized to release mRNA, which hybridized to the spatially barcoded capture probes, enabling the capture of gene expression data. Barcoded cDNA was synthesized on the slide from the captured mRNA, denatured, cleaved, and transferred into a PCR tube for amplification. The amplified cDNA was then used to generate standard next-generation sequencing (NGS) libraries. In brief, the amplified cDNA was enzymatically fragmented, purified, and size-selected. Sequencing adapters were ligated to each fragment, followed by sample index PCR. The resulting libraries were sequenced on a NovaSeq6000 SP flow cell with an average of 50,000 reads per probe, using a 28x10x10x90 read format.

## Analysis of visium data

**QC and cluster analysis.** FASTQ files were analyzed using Spaceranger v2.1.1 using Sus scrofa genome v11.1 downloaded from Ensembl (www.ensembl.org). Filtered feature matrices were then utilized for secondary analyses in R using a Seurat-based pipeline [59]. Briefly, the packages Seurat v4.4.0, SeuratObject v5.0.1, and SeuratWrappers v0.2.0, devtools 2.4.5, dplyr v1.1.4, harmony v1.2.0 and patchwork v1.2.0 were used for analysis. Individual imported filtered feature matrices underwent filtering for to only include cells with nFeatures falling above 200 and percent ribosomal reads lower than 20. Matrices were then merged, and individually normalized using SCTransform, regressing percent ribosomal reads, nFeatures, and nCounts. Variable features across the datasets were then obtained using SelectIntegrationFeatures and datasets were remerged using Merge_Seurat_List. Variable features were then set using VariableFeatures and dimensional reduction and cluster analysis performed using RunPCA on the top 50 dimensions. Cluster analysis was then perfomed using FindNeighbors (top 30 dimensions), FindClusters (resolution of 0.3), and RunUMAP (top 30 dimensions). Prior to subsequent analyses, where indicated, ALRA (RunALRA) mediated imputation of the data was performed as previously described. Marker identification was performed using the FindAllMarkers function using the ALRA assay, with a minimum fold change threshold of 0.25 and detected in a minimum of 0.25 of cells. For marker analyses the default Seurat adjusted *p*-value of $p<0.01$ (Bonferroni correction) was considered significant.

**Probabilistic modeling based identification of sTO clusters.** To identify the cluster identities in sTO, we utilized a previously established method to map single-cell data to visium spatial transcriptomics data [59,65]. Briefly, visium spatial transcriptomics data were first subsetted to only include structures known to contain trophoblasts, which reduces

non-specific signal. Visium data were then set to the SCT data, whereas sTO single-cell data were set to the integrated assay. Probabilistic transfer of labels was then performed using FindTransferAnchors and TransferData functions, using sTO data as a reference and subsetted visium data as the query. To identify shared markers between sTO and their corresponding tissue structures, marker lists for the SCT assay of sTO and the SCT assay of the visium data were generated using FindAllMarkers with a minimum fold change threshold of 0.25 and detected in a minimum of 0.25 of cells, with adjusted $p$-values of 0.01 considered significant. Individual sTO clusters and their corresponding tissue structure clusters were then filtered, and the lists inner joined to generate shared marker lists.

**CellChat analysis for visium.** Cellchat was performed using previously established methods using the packages CellChat and NMF [63,64]. Briefly, interaction inputs, complex inputs, cofactor inputs, and gene info inputs were downloaded from CellChatDB and cellchat objects were created using the ALRA assay. Overexpressed genes and interactions were then identified using the identifyOverExpressedGenes and identifyOverExpressedInteractions functions. Communication probabilities were then calculated using the computeCommunProb function. Interactions were corrected for both distance and cell-to-cell contact, with the maximum diffusion interaction/diffusion length being set to 250 μm and the cell spot center to spot center range being set to 100 μm. Communications were then filtered to include interactions including a minimum of 10 cells (filtercommunication function) and probabilities calculated using computeCommunProbPathway function. Aggregated networks were then calculated using aggregateNet. Patterning analysis was performed first by selecting a number of patterns using the selectK function, which was assigned based on when the which Cophenetic and Silhouette values dropped suddenly. Pattern plots were generated using the identifyCommunicationPatterns function. Additional plots were generated using standard methods as described in detailed scripts.

## Comparative cellchat analysis for visium and single cell

Briefly, visium spatial transcriptomics data and sTO single-cell datasets were subset to only include areolar trophoblasts and interface trophoblast populations. These populations were then combined into a single areola and interface trophoblast cluster for each dataset. Cellchat was then performed as described for sTO (see above). The only modification for this analysis was the interaction filtering was modified to a minimum of 1 cell to account for rare cell populations. Additionally, for patterning analyses, only pathways found in both datasets are shown. All dotplots shown using the ALRA assay.

## Supporting information

**S1 Fig.** A) UMAP split by sample (MFI). Dots indicate individual visium spots. **B)** Spatial Dimplot showing separation of maternal and fetal components. Representative maternal–fetal Interface shown. **C)** Spatial DimPlot separated by sample type and histologic structures showing the localization of UMAP cluster populations.
(S1_Fig.JPG)

**S2 Fig.** A–E) Spatial feature plots of known and novel markers of various trophoblast populations, split by sample (MFI). Various populations include A) Pan trophoblast markers, B) Interface trophoblast markers, C) Pan-Areola markers, D) Areola-2 markers, and E) Areola-1 markers.
(S2_Fig.JPG)

**S3 Fig.** A, B) Spatial feature plot showing prediction scores for each spot in the spatial dataset subsetted by sTO cluster. Color shown indicates probability that the classified sTO cluster is localized to a given position within the spatial dataset. A) Scoring for MFI025. B) Scoring for MFI094.
(S3_Fig.JPG)

**S4 Fig. Differential expression highlights the distinct functional niches of interface and areola-2 trophoblasts. A)** Volcano plot showing differentially expressed genes in interface-trophoblasts and areola-2 trophoblasts within the

Visium Spatial Transcriptomics data. Significant differences ($p < 0.01$, Fold-change $> 0.25$) are shown in blue. **B)** Volcano plot showing differentially expressed genes in interface-trophoblasts and areola-2 trophoblasts within the sTO single-cell transcriptomics data. Significant differences ($p < 0.01$, Fold-change $> 0.25$) are shown in red. **C, D)** Dotplots showing the expression of endosome-specific genes in both Visium (C) and sTO datasets (D). Plots showing Interface-trophoblast enriched GO-Terms as calculated by DAVID pathway analysis (significant difference $= p < 0.05$) for Visium spatial transcriptomics (C) and sTO single cell (D). Color of dots denotes average expression, whereas size represents percentage of cells in a cluster expressing the gene of interest.
(S4_Fig.JPG)

**S5 Fig.** A–F) Spatial feature plots showing genes that are enriched during the transition to Areola-2 trophoblasts and their expression pattern within our spatial transcriptomics dataset, split my sample (MFI). A) GRN, B) ITM2C, C) NUS1, D) NUPR1, E) SQSTM1, and F) ARF4.
(S5_Fig.JPG)

**S1 Movie. 3D image reconstruction of sTO grown in matrigel.**
(S1_Movie.AVI)

**S2 Movie. 3D image reconstruction of sTO grown in suspension.**
(S2_Movie.AVI)

**S1 Table. Single-cell marker analysis from sTO.**
(S1_Table.CSV)

**S2 Table. Visium marker analysis from the porcine maternal–fetal interface.**
(S2_Table.CSV)

**S3 Table. Shared markers between sTO cluster 0 and Visium interface trophoblasts.**
(S3_Table.CSV)

**S4 Table. Shared markers between sTO cluster 1 and Visium areola-2 trophoblasts.**
(S4_Table.CSV)

**S5 Table. Shared markers between sTO cluster 3 and Visium areola-2 trophoblasts.**
(S5_Table.CSV)

**S6 Table. Shared markers between sTO cluster 4 and Visium areola-1 trophoblasts.**
(S6_Table.CSV)

**S7 Table. GO-terms enriched in Visium interface-trophoblasts.**
(S7_Table.CSV)

**S8 Table. GO-terms enriched in Visium areola-2 trophoblasts.**
(S8_Table.CSV)

**S9 Table. GO-terms enriched in sTO interface-trophoblasts.**
(S9_Table.CSV)

**S10 Table. GO-terms enriched in sTO areola-2 trophoblasts.**
(S10_Table.CSV)

**S11 Table. Gene expression signatures across sTO differentiation identified via Slingshot.**
(S11_Table.CSV)

## Acknowledgments

We would like to thank the Duke Molecular Genomics Core facility for assistance with the generation of bulk RNA-sequencing data, Single-Cell Transcriptomics data, and Visium Spatial Transcriptomics data.

## Author contributions

**Conceptualization:** Cole R. McCutcheon, Allyson Caldwell, Liheng Yang, Elisa Crisci, Jonathan Alex Pasternak, Carolyn B. Coyne.

**Data curation:** Cole R. McCutcheon, Elisa Crisci, Jonathan Alex Pasternak, Carolyn B. Coyne.

**Formal analysis:** Cole R. McCutcheon, Liheng Yang, Carolyn B. Coyne.

**Funding acquisition:** Jonathan Alex Pasternak, Carolyn B. Coyne.

**Investigation:** Cole R. McCutcheon, Allyson Caldwell, Liheng Yang, Elisa Crisci, Jonathan Alex Pasternak, Carolyn B. Coyne.

**Methodology:** Cole R. McCutcheon, Allyson Caldwell, Liheng Yang, Elisa Crisci, Jonathan Alex Pasternak, Carolyn B. Coyne.

**Project administration:** Cole R. McCutcheon, Jonathan Alex Pasternak, Carolyn B. Coyne.

**Supervision:** Carolyn B. Coyne.

**Validation:** Cole R. McCutcheon, Allyson Caldwell, Carolyn B. Coyne.

**Visualization:** Cole R. McCutcheon, Allyson Caldwell, Carolyn B. Coyne.

**Writing – original draft:** Cole R. McCutcheon, Carolyn B. Coyne.

**Writing – review & editing:** Cole R. McCutcheon, Allyson Caldwell, Liheng Yang, Elisa Crisci, Jonathan Alex Pasternak, Carolyn B. Coyne.

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
