## [Editor Report · Decision Letter 0]

25 Nov 2024

Dear Dr Coyne,

Thank you for submitting your manuscript entitled "Defining Cellular Diversity at the Swine Maternal-Fetal Interface Using Spatial Transcriptomics and Organoids" for consideration as a Research Article by PLOS Biology.

Your manuscript has now been evaluated by the PLOS Biology editorial staff as well as by an academic editor with relevant expertise and I am writing to let you know that we would like to send your submission out for external peer review.

Once your full submission is complete, your paper will undergo a series of checks in preparation for peer review. After your manuscript has passed the checks it will be sent out for review. To provide the metadata for your submission, please Login to Editorial Manager (https://www.editorialmanager.com/pbiology) within two working days, i.e. by Nov 27 2024 11:59PM.

Kind regards,

Luke

Lucas Smith, Ph.D.

Senior Editor

PLOS Biology

lsmith@plos.org

---

## [Decision Letter · Decision Letter 1]

22 Jan 2025

Dear Dr Coyne,

Thank you for your patience while your manuscript "Defining Cellular Diversity at the Swine Maternal-Fetal Interface Using Spatial Transcriptomics and Organoids" was peer-reviewed at PLOS Biology. It has now been evaluated by the PLOS Biology editors, an Academic Editor with relevant expertise, and by several independent reviewers.

In light of the reviews, which you will find at the end of this email, we would like to invite you to revise the work to thoroughly address the reviewers' reports.

As you will see below, the reviewers agree that your study fills an important gap in knowledge and will be of interest to researchers working in this field. However, they each raise important concerns. The reviewers highlight that additional analyses is needed to support some of the conclusions, that further clarifications and justifications are needed for some of the analyses performed here, and importantly that the raw data needs to be made available. We think the reviewer comments will need to be thoroughly addressed before we can consider your study again for publication. I do note that in a couple of instances, the reviewers suggest that you cite specific studies in specific locations in your paper - and while we encourage you to consider these requests, we would make those specific changes optional.

Given the extent of revision needed, we cannot make a decision about publication until we have seen the revised manuscript and your response to the reviewers' comments. Your revised manuscript is likely to be sent for further evaluation by all or a subset of the reviewers.

**IMPORTANT - SUBMITTING YOUR REVISION**

*Re-submission Checklist*

*Published Peer Review*

*PLOS Data Policy*

*Blot and Gel Data Policy*

Sincerely,

Luke

Lucas Smith, Ph.D.

Senior Editor

PLOS Biology

lsmith@plos.org

REVIEWS:

Reviewer #1: In this manuscript, McCutcheon and colleagues describe the derivation of trophoblast organoids from term swine placentas, and they characterize them compared to mid-gestation placenta using bulk RNA-seq, single cell RNA-seq, and spatial transcriptomics.

The manuscript is overall well written and introduces a novel interesting model to investigate swine placental biology. It also provides novel spatial data on mid-gestation swine placentas, which add significant information on cell types and structural organization of the swine maternal/fetal interface. The manuscript is well written and follows a logical structure. The Methods contain enough information to understand the experimental procedures and bioinformatics analysis performed. Statistical analyses are performed appropriately. Dataset generated are publicly available and a user-friendly version of the Visium spatial transcriptomics dataset is made available for public enquiry. The research is solid and addresses fundamental questions, which are of interest to placental biologists as well as evolutionary biologists. A few issues have been found: please, find below comments to address before the manuscript can be considered for publication.

Major

- Figure 1H. As presented this figure does not necessarily support the conclusion that "sTOs were more similar to placental tissue than PTr2 cells…" (line 158-159). In fact, if looking at Dim 1, which counts for the highest percentage of changes among samples, PTR2 sits in between TO and placental samples. Dim 2 is the factor that separates PTR2 from the other samples. However, other dimensions are not shown that might further separate TO from placental samples in a similar manner as dim 2 for PRT2. Analysis like K-means or clustertree (or others) are required to sustain the conclusion above.

- Figure 1I. How were these genes selected? They seem to contain a mix of trophoblast and stromal markers. A clear labelling of the cell types these markers represent in the figure would help with the interpretation of the data. Moreover, the data do not fully support the conclusion that "sTOs also expressed canonical markers of swine trophoblasts from placental tissue, whereas PTr2 cells did not (Figure 1I)". Some placental genes are equally not expressed in both PTR2 and sTO. Others are expressed in sTO but not at the same level as placental samples. Moreover, sTO seems to express genes not found in placental samples, some of which are in common to PTR2 and some are unique to sTO. A better representation of these data might also help to support the conclusion that "Together, these data indicate that sTO are composed of fetal-derived swine trophoblasts." (line 163-164). At this point this conclusion is only weakly supported by the data. Considering that the original cell suspension likely included both trophoblast and non-trophoblast cells, including fetal stromal cells and maternal decidual cells, it would be important to strengthen evidence that the derived sTO only comprise trophoblast cells.

- Figure 2A. what is the base for choosing 5 clusters instead of a lower or higher number? The cell population is quite tight and there are no clear cluster boundaries, so the choice of cluster number seems arbitrary. The rest of the data in figure 2 also does not sustain the choice of 5 clusters. A rationale for this choice should be presented and backed up by data.

- Figure 2C. ELF3 and FGFR2 are not canonical trophoblast markers, and they are not highly expressed in the dataset (mean around 1-1.5). ELF5 is a trophoblast progenitor marker in human and mouse. In human, ELF3 is expressed in EVT and decidual cells. Literature suggests that FGFR2 is only expressed at low levels in normal swine maternal-fetal interface [Edwards et al 2011 Reprod Biol Endocr.]

- Figure 2F. Choice of trophoblast subtypes not supported by literature: SLC2A1 has low expression in swine trophoblast cells and highly expressed in maternal epithelial cells [Kramer et al. 2020]; SLC2A3 do not seem to be specific to areola cells [Kramer et al. 2020], lack of reference for ITGB3: based on Frank et al 2010 (BOR) it looks specific to areolas in mid/late-gestation.

- Figure 2G. Data do not fully support the conclusion that "these various sTO cell populations are distinctive lineages rather than contaminating cell populations". Data on KRT18 were already shown in Fig. 2C and expression levels of CTSB are lower from the middle part of the UMAP cloud to the bottom suggesting at least cell heterogeneity.

- Line 292-293 "Although these markers partially resolved the clusters, their leaky expression across multiple clusters limited our ability to accurately assign cellular identities.": would a lower number of clusters potentially help with cleaner signature? Or does this suggest that sTOs do not fully recapitulate in vivo conditions to allow specific sub-type cell formation but instead promote hybrid gene expression?

- Fig. 5D-E. Cluster 2 did not map to any trophoblast cluster. However, this analysis only included the placental cell of trophoblast origin. They might be non-trophoblast cells. Maybe interstitial? Do they cluster with any of the other placental cell types not included in this analysis?

- Fig. 2B and 5H show the same data with different colors but the proportion of cell by definition is the same so it is not introducing new information.

- Both sTO clusters 1 and 3 only share partial non-overlapping gene expression signature with placental areola 2 cells. This suggests that sTOs only partially recapitulate cell types in vivo.

- Supplemental Figure 4C-F. GO analysis data are either too small or at too low resolution to be read. Unfortunately, I was not able to verify the results presented in the text and its conclusions.

- Lines 364-368. Suppl. Fig. 4G indicates that endosomal genes are specific to areola-2 trophoblast cells. However, expression in sTOs clusters is not specific to the areola cells but expression is distributed between the areola-2 and ProlTroph with significant albeit lower expression in IntTroph cluster. This points again to the presence of hybrid cells in sTO posing a huge limitation to the use of sTO to study swine placental biology.

- The premise of figure 7 (lines 414-422) is to use CellChat to investigate maternal/fetal crosstalk using sTO. However, the authors claim sTO are only comprised of trophoblast (fetal) cells therefore by definition it is not possible to study maternal/fetal crosstalk using only this model. However, it is possible to address that question in the Visium dataset. Therefore, figure 7 tries to combine two fundamentally different questions: (1) what are the crosstalks in the maternal/fetal interface in porcine placenta? (Visium dataset); and (2) do sTOs recapitulate the crosstalk within the trophoblast populations observed in vivo? These two questions should be treated separately in the story and data from the two datasets should not be presented in parallel.

- Moreover, the Visium dataset suggests the highest number/strength of interaction occurs with maternal cells. Even when only addressing the trophoblast crosstalk in Fig. 7G-N, the presence of maternal information distracts from the trophoblast only data in the placental samples (which is the focus of the comparison with sTO). To address the second question, sub-setting the Visium dataset including only the trophoblast cells might be a better strategy to compare to sTO.

Minor:

- The initial cell suspension is likely composed of both trophoblast and non-trophoblast cells, likely fetal stromal cells and potentially maternal cells. While some data suggest the derived organoids are trophoblast in origin, it would be important to present some characterization of the original cell suspension to compare to the cell population in established organoids.

- Fig 4A. It would be useful to include the labelled maps (Fig.3D) for easily referencing the cell types in the different placentas. (also applies to Suppl. Fig 2)

- Fig 5C is the same as 2A.

- Fig 5D-E. It would be useful to include the labelled maps (Fig.3D) for easily referencing the cell types in the different placentas.

- Line 387. Supplemental Figure 7 not provided. Is this actually suppl. Fig 5?

- Figure 6A. text indicates that the trajectory analysis was applied after sub-setting the dataset to include only ProlTroph, IntTroph, and Areola-2 cells. However, the top part of the figure shows all the clusters, and it seems contradictory.

- Fig. 6B. too many genes on the side of the heatmap makes it difficult to read and find any.

- Suppl. Figure 6. The panels are not referenced in order in the text

- Supplemental Fig. 6. The legend does not clarify what the labels on the side of the heatmaps represent: general signaling pathways or genes? Shouldn't incoming patterns be represented mainly by receptors?

- Line 452-454. This is a circular argument. The spatial data was used to predict the ligand-receptor interaction so it cannot be used also to validate the CellChat findings.

- Fig. 7N. genes too squeezed and not clearly legible even when zoomed in

- Lines 454-464/Fig 7I-J-M-N. Text should present data in order in the figure. GNR first, MK afterwards.

- Line 605. "within" repeated

Reviewer #2: This manuscript addresses a significant gap in placental biology by leveraging spatial transcriptomics and organoid models to explore the cellular heterogeneity of the porcine maternal-fetal interface. The study is timely and relevant, given the growing interest in understanding placental development across species. However, issues related to data and code availability, methodological clarity, statistic analysis, and citation of relevant literature need to be addressed before publication. Below, I provide detailed comments, organized by section, with recommendations to enhance the manuscript's rigor and transparency.

While the study is important, the data is not available for review and Github code is empty. As a standard practice, the author needs to make the data and code available for review. In addition, the manuscript lacks sufficient detail regarding data analysis pipelines, making it challenging to evaluate the rigor of the findings. Importantly, the authors did not reference a closely related paper "Cross-Species Insights into Trophoblast Invasion During Placentation Governed by Immune-Featured Trophoblast Cells," which provides valuable insights into early pig placentation at the single-cell level.

Including this reference and contextualizing the findings would strengthen the manuscript's contributions.

I recommend a revision for this manuscript and I would be delighted to read the revised manuscript.

Data and code availability

It is unusual to submit a single-cell atlas manuscript while the data and codes are not available to review, which is not the standard practice. Please check with the manuscript submission policy of the journal.

Bulk RNA-seq Analysis

Lines 782-784: It is unclear which analyses were performed using DESeq2. As detailed in the DESeq2 manual (https://bioconductor.org/packages/3.21/bioc/vignettes/DESeq2/inst/doc/DESeq2.html#why-un-normalized-counts), DESeq2 expects raw, un-normalized counts as input, not RPKMs. The authors should clarify their input data and ensure that the pipeline aligns with best practices.

scRNA-seq analysis

Line 850-851. The manuscript does not specify the adjusted p-value threshold used. Including this information is critical for evaluating the robustness of differential expression analyses.

Differential Expression and GO-Term Analysis

Line 868. The authors used a p-value < 0.01 threshold for differential expression. However, when performing multiple hypothesis testing, it is recommended to use adjusted p-values to control for false discovery rates. Please justify this choice or revise the analysis using corrected p-values.

CellChat Analysis for scRNA-seq

Line 874-888. It is unclear how CellChat was applied to pig data, as the software defaults to human and mouse gene sets. Did the authors identify human-pig one-to-one orthologs and use only this subset of genes for the analysis? Detailed methods should be provided, as the GitHub links mentioned in the manuscript are empty or not informative to replicate the analysis.

Visium analysis

Lines 912-915. The decision to merge, integrate, and then re-merge Visium datasets appears methodologically incorrect. Integration already accounts for batch effects. The authors should justify this approach.

Before integration, did the authors run PrepSCTIntegration? This is a required step for SCT-based integration.

Line 914. Which method was used for the integration? SelectIntegrationFeatures alone does not complete the integration process.

Line 915. After integration, did the authors set the default assay to "integrated"? Additionally, why were new variable features obtained post-integration? This requires clarification.

Line 922. The manuscript does not specify the adjusted p-value threshold used. This information is essential for evaluating the rigor of the analysis.

Lines 932-935. Identifying markers for the integrated assay of spatial transcriptomics (sTO) is methodologically incorrect. For differential expression analysis, the assay must be set to "RNA." Please also clarify the adjusted p-value threshold used in this analysis.

CellChat Analysis for Visium

Line 938-952. Similar to the scRNA-seq analysis, it is unclear how CellChat was applied to pig data. If the authors used human-pig orthologs, they should provide details about the identification process and the subset of genes used. As previously noted, the reported GitHub links are empty or not sufficiently detailed to replicate the analysis. Ensuring reproducibility is crucial for validating the results.

Reviewer #3: McCutcheon et al have described the isolation, propogation, and identification of swine trophoblast organoids and referenced the cell types back to the swine maternal-fetal-interface utilizing spatial transcriptomics. The paper is very novel in that this is the first time swine organoids have been isolated and it is the first visium spatial transcriptomic profile done to determine the transcriptional profile of the swine maternal-fetal-interface. It was also shown that previously utilized canonical markers of different trophoblast populations identified by scRNAseq were not sufficient to identify the cell clusters, therefore spatial transcriptomics was utilized to pull new markers of those trophoblast cell types. The paper concludes by analyzing the cell-cell communication networks between the trophoblast and maternal cell types.

Introduction-

Please cite Arutyunyan et al Nature 2023 particularly in the first sentence of line 79. Also in line 87, Arutyunyan and Shannon et al Dev Cell 2024. Line 110, cite Haider et al 2018.

Figure 1-

A-D) How did you assess that you were able to isolate purely trophoblast stem cells? Please include FACS markers, IFs, that trophoblast markers are present. I realize not all human antibodies will work with pig, but please try! I don't think it's enough to say, they are proliferating as spherical organoids, therefore they are trophoblast stem cells. Especially considering your bulk seq in park I-

I) Many of the transcriptional profiles appear opposite of term placenta, and about half of it is definitely opposite of PTR2 (which is the point, I realize, but what trophoblast type does PTR2 represent?). The only regions that look like TOs match term placenta is MMP24-SOX2, WFDC2-PLET1. Could the canonical genes of interest that match placental tissue be highlighted?

H) could a maternal epithelial cell type be included in this analysis to demonstrate distinction from this cell type?

G) The organoids are grown in matrigel droplets which do not replicate chorionic tissue architecture, and then grown in suspension, this does appear better in terms of the function of the trophoblasts and the cellular orientations, why wasn't this chosen for scRNAseq? This question is related to figure 2-

Can columnar morphology be observed in any of the organoids that indicates not just transcriptional alignment with areola, but in cell morphology as well?

Figure 2- Why were the matrigel-embedded sTOs used for scRNAseq? This is never justified based on the observations of the previous figure. To me, it seems that the previous figure (1) shows that the suspension organoids replicate more relevant in vivo orientation? So, why were matrigel-embedded TOs used?

C) What about other traditional mouse/human trophoblast markers like GATA2/3 and TEAD4?

C/D) There is no stromal contamination based on this data, but what about maternal epithelial contamination? Please include markers of endoderm (assuming this is the lineage of the swine endometrium) to compare as well.

Figure 6- I have reservations that the differentiation progression is indeed a linear ProfTroph -> IntTroph -> Areola-2. Unless this is a well known progression that I am unaware of, your data could also indicate a bifurcating differentiation pattern from ProfTroph to IntTroph and ProfTroph to Areola-2 separately. I'm unfamiliar with this Trajectory inference analysis, but is a bifurcating pattern a possible output? If it is only programmed to recognize linear progressions, then the software will find a way to fit a lineage progression. I would like to see either a change in language in the text to the possibility that there could be a bifurcating pattern, but there is not software to recognize this, or the authors may have to perform a timecourse of multiple samples to discover whether the two lineages arise temporally simultaneously (perhaps indicating a bifurcation) or sequentially (indicating a linear progression). To me, your data as presented indicates a bifurcation, and this also makes more sense when in previous figures, the authors describe very different functions of each lineage. Like other figures, I would like to see the authors spatially locate the transcripts within the organoids. This could be done in a timecourse, and does not need to be done with expensive multiple scRNAseq samples. Using your markers, identify with qPCR or HCR-RNAfish (or something similar) where and when these transcripts arise. Or, bulk seq multiple time points?

Questions beyond the figures-

Given your vast transcriptome data, could you alter culture conditions such that areolar cells could be enriched or matured?

Can you spatially resolve where the trophoblast populations reside within the organoids? How does it change when the organoids are cultured in suspension?

---

## [Decision Letter · Decision Letter 2]

25 Jun 2025

Dear Dr Coyne,

Thank you for your patience while we considered your revised manuscript "Defining Cellular Diversity at the Swine Maternal-Fetal Interface Using Spatial Transcriptomics and Organoids" for publication as a Research Article at PLOS Biology. This revised version of your manuscript has been evaluated by the PLOS Biology editors, the Academic Editor and the original reviewers.

All three reviewers and our Academic Editor have indicated that they are fully satisfied by the revision. However, before we can editorially accept your study, we need you to address a few data and other policy-related requests in a last, short, revision. These are enumerated below.

**IMPORTANT: Please address the following editorial requests**

1) ARTICLE TYPE: After some discussion within the team, we think that your paper would be best suited for publication as a "Methods and Resources" article. We therefore ask that you change the article type, in our editorial manager system, accordingly. This should not require any changes to the format or presentation of the manuscript.

2) DATA/CODE AVAILABILITY: I see that you have provided a detailed data availability statement, indicating where your code and data can be found, in the text of your manuscript (thank you!).

>>Please move that statement to the relevant "data availability statement" section in our editorial manager system. That is the version that will be published with your paper.

>>Please add a sentence to each figure legend, pointing readers to the relevant underlying data. For example, you can add the sentence "the data underlying this figure can be found on the Gene Expression Omnibus (GSE296975)

3) SUPPLEMENTAL FIGURES: I see in your manuscript that you have 5 supplemental figures - but looking through your PDF and file inventory, I could not seem to find these anywhere. Sorry if I missed something obvious here, but please do double check that you have uploaded all the relevant files for your paper.

4) CODE: Thank you for providing your code on GitHub. While this is great, please note that we cannot accept sole deposition of code in GitHub, as this could be changed after publication. We therefore ask that you archive this version of your publicly available GitHub code to Zenodo. Once you do this, it will generate a DOI number, which you will need to provide in the Data Availability Statement (you are welcome to also provide the GitHub access information). See the process for doing this here: https://docs.github.com/en/repositories/archiving-a-github-repository/referencing-and-citing-content

5) ETHICS STATEMENT: Thank you for including an ethics statement in your methods section. Please update this to include the specific national or international regulations/guidelines to which your animal care and use protocol adhered. Please note that institutional or accreditation organization guidelines (such as AAALAC) do not meet this requirement.

We expect to receive your revised manuscript within two weeks.

*Published Peer Review History*

*Press*

Sincerely,

Luke

Lucas Smith, Ph.D.

Senior Editor

lsmith@plos.org

PLOS Biology

Reviewer remarks:

Reviewer #1, Francesca Soncin [note: reviewer 1 has signed their review]: I would like to thank the authors for carefully considering the previous comments and for their thoughtful responses. The manuscript was satisfactorily updated and I have no further comments or concerns. This was a tremendous effort and inspiring work.

Reviewer #2: The authors have answered my questions.

Reviewer #3 [note: reviewer 3 did review the revision, but provided no additional comments for the authors]:

---

## [Editor Report · Decision Letter 3]

7 Jul 2025

Dear Dr Coyne,

Thank you for the submission of your revised Methods and Resources "Defining Cellular Diversity at the Swine Maternal-Fetal Interface Using Spatial Transcriptomics and Organoids" for publication in PLOS Biology and thank you for addressing our editorial requests in this revision. On behalf of my colleagues and the Academic Editor, Carmen J Williams, I am pleased to say that we can in principle accept your manuscript for publication, provided you address any remaining formatting and reporting issues. These will be detailed in an email you should receive within 2-3 business days from our colleagues in the journal operations team; no action is required from you until then. Please note that we will not be able to formally accept your manuscript and schedule it for publication until you have completed any requested changes.

**IMPORTANT: I ended up pasting the data availability statement that was included in your manuscript into the relevant section of our editorial manager system, as this will be the version that is published with your paper. Please do take a moment to make sure that the updated statement looks OK after this change.

PRESS

Sincerely, 

Lucas Smith, Ph.D.

Senior Editor

PLOS Biology

lsmith@plos.org